# Coresets for Regressions with Panel Data

**Lingxiao Huang**
Huawei

**K. Sudhir**
Yale University

**Nisheeth K. Vishnoi**
Yale University

## Abstract

This paper introduces the problem of coresets for regression problems to panel data settings. We first define coresets for several variants of regression problems with panel data and then present efficient algorithms to construct coresets of size that depend polynomially on $1/\varepsilon$ (where $\varepsilon$ is the error parameter) and the number of regression parameters – independent of the number of individuals in the panel data or the time units each individual is observed for. Our approach is based on the Feldman-Langberg framework in which a key step is to upper bound the "total sensitivity" that is roughly the sum of maximum influences of all individual-time pairs taken over all possible choices of regression parameters. Empirically, we assess our approach with a synthetic and a real-world datasets; the coreset sizes constructed using our approach are much smaller than the full dataset and coresets indeed accelerate the running time of computing the regression objective.

## 1 Introduction

Panel data, represented as $X \in \mathbb{R}^{N \times T \times d}$ and $Y \in \mathbb{R}^{N \times T}$ where $N$ is the number of entities/individuals, $T$ is the number of time periods and $d$ is the number of features is widely used in statistics and applied machine learning. Such data track features of a cross-section of entities (e.g., customers) longitudinally over time. Such data are widely preferred in supervised machine learning for more accurate prediction and unbiased inference of relationships between variables relative to cross-sectional data (where each entity is observed only once) [28, 6].

The most common method for inferring relationships between variables using observational data involves solving regression problems on panel data. The main difference between regression on panel data when compared to cross-sectional data is that there may exist correlations within observations associated with entities over time periods. Consequently, the regression problem for panel data is the following optimization problem over regression variables $\beta \in \mathbb{R}^d$ and the covariance matrix $\Omega$ that is induced by the abovementioned correlations: $\min_{\beta \in \mathbb{R}^d, \Omega \in \mathbb{R}^{T \times T}} \sum_{i \in [N]} (y_i - X_i \beta)^\top \Omega^{-1} (y_i - X_i \beta)$. Here $X_i \in \mathbb{R}^{T \times d}$ denotes the observation matrix of entity $i$ whose $t$-th row is $x_{it}$ and $\Omega$ is constrained to have largest eigenvalue at most 1 where $\Omega_{tt'}$ represents the correlation between time periods $t$ and $t'$. This regression model is motivated by the random effects model (Eq. (1) and Appendix A), common in the panel data literature [27, 24, 23]. A common way to define the correlation between observations is an autocorrelation structure $\mathsf{AR}(q)$ [25, 35] whose covariance matrix $\Omega$ is induced by a vector $\rho \in \mathbb{R}^q$ (integer $q \geq 1$). This type of correlation results in the generalized least-squares estimator (GLSE), where the parameter space is $\mathcal{P} = R^{d+q}$.

As the ability to track entities on various features in real-time has grown, panel datasets have grown massively in size. However, the size of these datasets limits the ability to apply standard learning algorithms due to space and time constraints. Further, organizations owning data may want to share only a subset of data with others seeking to gain insights to mitigate privacy or intellectual property related risks. Hence, a question arises: *can we construct a smaller subset of the panel data on which we can solve the regression problems with performance guarantees that are close enough to those obtained when working with the complete dataset?*

One approach to this problem is to appeal to the theory of "coresets." Coresets, proposed in [1], are weighted subsets of the data that allow for fast approximate inference for a large dataset by solving the problem on the smaller coreset. Coresets have been developed for a variety of unsupervised and supervised learning problems; for a survey, see [43]. But, thus, far coresets have been developed only for $\ell_2$-regression cross-sectional data [18, 36, 8, 15, 33]; no coresets have been developed for regressions on panel data – an important limitation, given their widespread use and advantages.

Roughly, a coreset for cross-sectional data is a weighted subset of observations associated with entities that approximates the regression objective for every possible choice of regression parameters. An idea, thus, is to construct a coreset for each time period (cross-section) and output their union as a coreset for panel data. However, this union contains at least $T$ observations which is undesirable since $T$ can be large. Further, due to the covariance matrix $\Omega$, it is not obvious how to use this union to approximately compute regression objectives. With panel data, one needs to consider both how to sample entities, and within each entity how to sample observations across time. Moreover, we also need to define how to compute regression objectives on such a coreset consisting of entity-time pairs.

**Our contributions.** We initiate the study of coresets for versions of $\ell_2$-regression with panel data, including the ordinary least-squares estimator (OLSE; Definition 2.2), the generalized least-squares estimator (GLSE; Definition 2.3), and a clustering extension of GLSE (GLSE$_k$; Definition 2.4) in which all entities are partitioned into $k$ clusters and each cluster shares the same regression parameters.

Overall, we formulate the definitions of coresets and propose efficient construction of $\varepsilon$-coresets of sizes independent of $N$ and $T$. Our key contributions are: **(a)** We give a novel formulation of coresets for GLSE (Definition 3.1) and GLSE$_k$ (Definition 3.2). We represent the regression objective of GLSE as the sum of $NT$ sub-functions w.r.t. entity-time pairs, which enables us to define coresets similar to the case of cross-sectional data. For GLSE$_k$, the regression objective cannot be similarly decomposed due to the $\min$ operations in Definition 2.4. To deal with this issue, we define the regression objective on a coreset $S$ by including $\min$ operations. **(b)** Our coreset for OLSE is of size $O(\min\{\varepsilon^{-2}d, d^2\})$ (Theorems C.1 and C.2), based on a reduction to coreset for $\ell_2$-regression with cross-sectional data. **(c)** Our coreset for GLSE consists of at most $\tilde{O}(\varepsilon^{-2}\max\{q^4d^2, q^3d^3\})$ points (Theorem 4.1), independent of $N$ and $T$ as desired. **(d)** Our coreset for GLSE$_k$ is of size $\text{poly}(M, k, q, d, 1/\varepsilon)$ (Theorem 5.2) where $M$ upper bounds the gap between the maximum individual regression objective of OLSE and the minimum one (Definition 5.1). We provide a matching lower bound $\Omega(N)$ (Theorem 5.4) for $k, q, d \leq 2$, indicating that the coreset size should contain additional factors than $k, q, d, 1/\varepsilon$, justifying the $M$-bounded assumption.

Our coresets for GLSE/GLSE$_k$ leverage the Feldman-Langberg (FL) framework [21] (Algorithms 1 and 2). The $\rho$ variables make the objective function of GLSE non-convex in contrast to the cross-sectional data setting where objective functions are convex. Thus, bounding the "sensitivity" (Lemma 4.4) of each entity-time pair for GLSE, which is a key step in coreset construction using the FL framework, becomes significantly difficult. We handle this by upper-bounding the maximum effect of $\rho$, based on the observation that the gap between the regression objectives of GLSE and OLSE with respect to the same $\beta \in \mathbb{R}^d$ is always constant, which enables us to reduce the problem to the cross-sectional setting. For GLSE$_k$, a key difficulty is that the clustering centers are *subspaces* induced by regression vectors, instead of *points* as in Gaussian mixture models or $k$-means. Hence, it is unclear how GLSE$_k$ can be reduced to projective clustering used in Gaussian mixture models; see [20]. To bypass this, we consider observation vectors of an individual as one entity and design a two-staged framework in which the first stage selects a subset of individuals that captures the $\min$ operations in the objective function and the second stage applies our coreset construction for GLSE on each selected individuals. As in the case of GLSE, bounding the "sensitivity" (Lemma E.4) of each entity for GLSE$_k$ is a key step at the first stage. Towards this, we relate the total sensitivity of entities to a certain "flexibility" (Lemma E.3) of each individual regression objective which is, in turn, shown to be controlled by the $M$-bounded assumption (Definition 5.1).

We implement our GLSE coreset construction algorithm and test it on synthetic and real-world datasets while varying $\varepsilon$. Our coresets perform well relative to uniform samples on multiple datasets with different generative distributions. Importantly, the relative performance is robust and better on datasets with outliers. The maximum empirical error of our coresets is always below the guaranteed $\varepsilon$ unlike with uniform samples. Further, for comparable levels of empircal error, our coresets perform much better than uniform sampling in terms of sample size and coreset construction speed.

## 1.1 Related work

With panel data, depending on different generative models, there exist several ways to define $\ell_2$-regression [27, 24, 23], including the pooled model, the fixed effects model, the random effects model, and the random parameters model. In this paper, we consider the random effects model (Equation (1)) since the number of parameters is independent of $N$ and $T$ (see Section A for more discussion).

For cross-sectional data, there is more than a decade of extensive work on coresets for regression; e.g., $\ell_2$-regression [18, 36, 8, 15, 33], $\ell_1$-regression [11, 47, 12], generalized linear models [31, 40] and logistic regression [44, 31, 42, 49]. The most relevant for our paper is $\ell_2$-regression (least-squares regression), which admits an $\varepsilon$-coreset of size $O(d/\varepsilon^2)$ [8] and an accurate coreset of size $O(d^2)$ [33].

With cross-sectional data, coresets have been developed for a large family of problems in machine learning and statistics, including clustering [21, 22, 30], mixture model [37], low rank approximation [16], kernel regression [54] and logistic regression [42]. We refer interested readers to recent surveys [41, 19]. It is interesting to investigate whether these results can be generalized to panel data.

There exist other variants of regression sketches beyond coreset, including weighted low rank approximation [13], row sampling [17], and subspace embedding [47, 39]. These methods mainly focus on the cross-sectional setting. It is interesting to investigate whether they can be adapted to the panel data setting that with an additional covariance matrix.

## 2 $\ell_2$-regression with panel data

We consider the following generative model of $\ell_2$-regression: for $(i, t) \in [N] \times [T]$,

$$y_{it} = x_{it}^\top \beta_i + e_{it}, \tag{1}$$

where $\beta_i \in \mathbb{R}^d$ and $e_{it} \in \mathbb{R}$ is the error term drawn from a normal distribution. Sometimes, we may include an additional entity or individual specified effect $\alpha_i \in \mathbb{R}$ so that the outcome can be represented by $y_{it} = x_{it}^\top \beta_i + \alpha_i + e_{it}$. This is equivalent to Equation (1) by appending an additional constant feature to each observation $x_{it}$.

**Remark 2.1** *Sometimes, we may not observe individuals for all time periods, i.e., some observation vectors $x_{it}$ and their corresponding outcomes $y_{it}$ are missing. One way to handle this is to regard those missing individual-time pairs as $(x_{it}, y_{it}) = (0, 0)$. Then, for any vector $\beta \in \mathbb{R}^d$, we have $y_{it} - x_{it}^\top \beta = 0$ for each missing individual-time pairs.*

As in the case of cross-sectional data, we assume there is no correlation between individuals. Using this assumption, the $\ell_2$-regression function can be represented as follows: for any regression parameters $\zeta \in \mathcal{P}$ ($\mathcal{P}$ is the parameter space), $\psi(\zeta) = \sum_{i \in [N]} \psi_i(\zeta)$, where $\psi_i$ is the individual regression function. Depending on whether there is correlation within individuals and whether $\beta_i$ is unique, there are several variants of $\psi_i$. The simplest setting is when all $\beta_i$s are the same, say $\beta_i = \beta$, and there is no correlation within individuals. This setting results in the ordinary least-squares estimator (OLSE); summarized in the following definition.

**Definition 2.2 (Ordinary least-squares estimator (OLSE))** *For an ordinary least-squares estimator (OLSE), the parameter space is $\mathbb{R}^d$ and for any $\beta \in \mathbb{R}^d$ the individual objective function is*

$$\psi_i^{(O)}(\beta) := \sum_{t \in [T]} \psi_{it}^{(O)}(\beta) = \sum_{t \in [T]} (y_{it} - x_{it}^\top \beta)^2.$$

Consider the case when $\beta_i$ are the same but there may be correlations between time periods within individuals. A common way to define the correlation is called autocorrelation $\mathsf{AR}(q)$ [25, 35], in which there exists $\rho \in B^q$, where $q \geq 1$ is an integer and $B^q = \{x \in \mathbb{R}^q : \|x\|_2 < 1\}$, such that

$$e_{it} = \sum_{a=1}^{\min\{t-1, q\}} \rho_a e_{i, t-a} + N(0, 1). \tag{2}$$

This autocorrelation results in the generalized least-squares estimator (GLSE).

**Definition 2.3 (Generalized least-squares estimator (GLSE))** *For a generalized least-squares estimator (GLSE) with $\mathsf{AR}(q)$ (integer $q \geq 1$), the parameter space is $\mathbb{R}^d \times B^q$ and for any $\zeta = (\beta, \rho) \in \mathbb{R}^d \times B^q$ the individual objective function is $\psi_i^{(G,q)}(\zeta) := \sum_{t \in [T]} \psi_{it}^{(G,q)}(\zeta)$ equal to*

$$(1 - \|\rho\|_2^2)(y_{i1} - x_{i1}^\top \beta)^2 + \sum_{t=2}^T \left( (y_{it} - x_{it}^\top \beta) - \sum_{j=1}^{\min\{t-1, q\}} \rho_j (y_{i, t-j} - x_{i, t-j}^\top \beta) \right)^2.$$

The main difference from OLSE is that a sub-function $\psi_{it}^{(G,q)}$ is not only determined by a single observation $(x_{it}, y_{it})$; instead, the objective of $\psi_{it}^{(G,q)}$ may be decided by up to $q+1$ contiguous observations $(x_{i,\max\{1,t-q\}}, y_{i,\max\{1,t-q\}}), \dots, (x_{it}, y_{it})$.

Motivated by $k$-means clustering [48], we also consider a generalized setting of GLSE, called $\text{GLSE}_k$ ($k \geq 1$ is an integer), in which all individuals are partitioned into $k$ clusters and each cluster corresponds to the same regression parameters with respect to some GLSE.

**Definition 2.4 ($\text{GLSE}_k$: an extention of GLSE)** *Let $k, q \geq 1$ be integers. For a $\text{GLSE}_k$, the parameter space is $\left(\mathbb{R}^d \times B^q\right)^k$ and for any $\zeta = (\beta^{(1)}, \dots, \beta^{(k)}, \rho^{(1)}, \dots, \rho^{(k)}) \in \left(\mathbb{R}^d \times B^q\right)^k$ the individual objective function is $\psi_i^{(G,q,k)}(\zeta) := \min_{l \in [k]} \psi_i^{(G,q)}(\beta^{(l)}, \rho^{(l)})$.*

$\text{GLSE}_k$ is a basic problem with applications in many real-world fields; as accounting for *unobserved heterogeneity* in panel regressions is critical for unbiased estimates [3, 26]. Note that each individual selects regression parameters $(\beta^{(l)}, \rho^{(l)})$ ($l \in [k]$) that minimizes its individual regression objective for GLSE. Note that $\text{GLSE}_1$ is exactly GLSE. Also note that $\text{GLSE}_k$ can be regarded as a generalized version of clustered linear regression [4], in which there is no correlation within individuals.

# 3   Our coreset definitions

In this section, we show how to define coresets for regression on panel data, including GLSE and $\text{GLSE}_k$. Due to the additional autocorrelation parameters, it is not straightforward to define coresets for GLSE as in the cross-sectional setting. One way is to consider all observations of an individual as an indivisible group and select a collection of individuals as a coreset. However, this construction results in a coreset of size depending on $T$, which violates the expectation that the coreset size should be independent of $N$ and $T$. By Definition 2.3, we know that the objective function $\psi^{(G,q)}$ can be represented as the summation of $NT$ sub-functions $\psi_{it}^{(G,q)}$. This motivated the following definition.

**Definition 3.1 (Coresets for GLSE)** *Given a panel dataset $X \in \mathbb{R}^{N \times T \times d}$ and $Y \in \mathbb{R}^{N \times T}$, a constant $\varepsilon \in (0,1)$, integer $q \geq 1$, and parameter space $\mathcal{P}$, an $\varepsilon$-coreset for GLSE is a weighted set $S \subseteq [N] \times [T]$ together with a weight function $w : S \to \mathbb{R}_{\geq 0}$ such that for any $\zeta = (\beta, \rho) \in \mathcal{P}$,*

$$\psi_S^{(G,q)}(\zeta) := \sum_{(i,t) \in S} w(i,t) \cdot \psi_{it}^{(G,q)}(\zeta) \in (1 \pm \varepsilon) \cdot \psi^{(G,q)}(\zeta).$$

Note that the number of points in this coreset $S$ is at most $(q+1) \cdot |S|$. Specifically, for OLSE, the parameter space is $\mathbb{R}^d$ since $q = 0$, and hence is a special case of the above definition. Also note that this definition can be derived from the coreset definition from [21, 9]; see Section B.1 for details.

Due to the $\min$ operations in Definition 2.4, the objective function $\psi^{(G,q,k)}$ can only be decomposed into sub-functions $\psi_i^{(G,q,k)}$ instead of individual-time pairs. Hence, the first idea is to select a sub-collection of $\psi_i^{(G,q,k)}$ to estimate the full function $\psi^{(G,q,k)}$. However, each sub-function $\psi_i^{(G,q,k)}$ is computed by $T$ observations and the resulting coreset size should contain a factor $T$. To avoid the size dependence of $T$, the intuition is to further select a subset of time periods to estimate $\psi_i^{(G,q,k)}$. Given $S \subseteq [N] \times [T]$, we denote $I_S := \{i \in [N] : \exists t \in [T], s.t., (i,t) \in S\}$ as the collection of individuals that appear in $S$. Moreover, for each $i \in I_S$, we denote $J_{S,i} := \{t \in [T] : (i,t) \in S\}$ to be the collection of observations for individual $i$ in $S$.

**Definition 3.2 (Coresets for $\text{GLSE}_k$)** *Given a panel dataset $X \in \mathbb{R}^{N \times T \times d}$ and $Y \in \mathbb{R}^{N \times T}$, constant $\varepsilon \in (0,1)$, integer $k, q \geq 1$, and parameter space $\mathcal{P}^k$, an $\varepsilon$-coreset for $\text{GLSE}_k$ is a weighted set $S \subseteq [N] \times [T]$ together with a weight function $w : S \to \mathbb{R}_{\geq 0}$ such that for any $\zeta = (\beta^{(1)}, \dots, \beta^{(k)}, \rho^{(1)}, \dots, \rho^{(k)}) \in \mathcal{P}^k$,*

$$\psi_S^{(G,q,k)}(\zeta) := \sum_{i \in I_S} \min_{l \in [k]} \sum_{t \in J_{S,i}} w(i,t) \cdot \psi_{it}^{(G,q)}(\beta^{(l)}, \rho^{(l)}) \in (1 \pm \varepsilon) \cdot \psi^{(G,q,k)}(\zeta).$$

The key is to incorporate $\min$ operations in the computation function $\psi_S^{(G,q,k)}$ over the coreset. Similar to GLSE, the number of points in such a coreset $S$ is at most $(q+1) \cdot |S|$.

# 4  Coresets for GLSE

In this section, we show how to construct coresets for GLSE. Due to space limitations, we omit many details to Section D. We let the parameter space be $\mathcal{P}_\lambda = \mathbb{R}^d \times B_{1-\lambda}^q$ for some constant $\lambda \in (0, 1)$ where $B_{1-\lambda}^q = \left\{ \rho \in \mathbb{R}^q : \|\rho\|_2^2 \leq 1 - \lambda \right\}$. The assumption of the parameter space $B_{1-\lambda}^q$ for $\rho$ is based on the fact that $\|\rho\|_2^2 < 1$ ($\lambda \to 0$) is a stationary condition for $\mathsf{AR}(q)$ [35].

**Theorem 4.1 (Coresets for GLSE)** *There exists a randomized algorithm that, for a given panel dataset $X \in \mathbb{R}^{N \times T \times d}$ and $Y \in \mathbb{R}^{N \times T}$, constants $\varepsilon, \delta, \lambda \in (0, 1)$ and integer $q \geq 1$, with probability at least $1 - \delta$, constructs an $\varepsilon$-coreset for GLSE of size $O\left( \varepsilon^{-2} \lambda^{-1} q d \left( \max \left\{ q^2 d, q d^2 \right\} \cdot \log \frac{d}{\lambda} + \log \frac{1}{\delta} \right) \right)$ and runs in time $O(NTq + NTd^2)$.*

Note that the coreset in the above theorem contains at most $(q + 1) \cdot O\left( \varepsilon^{-2} \lambda^{-1} q d \left( \max \left\{ q^2 d, q d^2 \right\} \cdot \log \frac{d}{\lambda} + \log \frac{1}{\delta} \right) \right)$ points $(x_{it}, y_{it})$, which is independent of both $N$ and $T$. Also note that if both $\lambda$ and $\delta$ are away from 0, e.g., $\lambda = \delta = 0.1$ the number of points in the coreset can be further simplified: $O\left( \varepsilon^{-2} \max \left\{ q^4 d^2, q^3 d^3 \right\} \cdot \log d \right) = \operatorname{poly}(q, d, 1/\varepsilon)$.

## 4.1  Algorithm for Theorem 4.1

We summarize the algorithm of Theorem 4.1 in Algorithm 1, which takes a panel dataset $(X, Y)$ as input and outputs a coreset $S$ of individual-time pairs. The main idea is to use importance sampling (Lines 6-7) leveraging the Feldman-Langberg (FL) framework [21, 9]. The key new step appears in Line 5, which computes a sensitivity function $s$ for GLSE that defines the sampling distribution. Also note that the construction of $s$ is based on another function $s^{(O)}$ (Line 4), which is actually a sensitivity function for OLSE that has been studied in the literature [8].

---

**Algorithm 1:** CGLSE: Coreset construction of GLSE

**Input:** $X \in \mathbb{R}^{N \times T \times d}$, $Y \in \mathbb{R}^{N \times T}$, constant $\varepsilon, \delta, \lambda \in (0, 1)$, integer $q \geq 1$ and parameter space $\mathcal{P}_\lambda$.
**Output:** a subset $S \subseteq [N] \times [T]$ together with a weight function $w : S \to \mathbb{R}_{\geq 0}$.

1: $M \leftarrow O\left( \varepsilon^{-2} \lambda^{-1} q d \left( \max \left\{ q^2 d, q d^2 \right\} \cdot \log \frac{d}{\lambda} + \log \frac{1}{\delta} \right) \right)$.
2: Let matrix $Z \in \mathbb{R}^{NT \times (d+1)}$ be whose $(iT - T + t)$-th row is $z_{it} = (x_{it}, y_{it}) \in \mathbb{R}^{d+1}$ for $(i, t) \in [N] \times [T]$.
3: Compute $A \subseteq \mathbb{R}^{NT \times d'}$ whose columns form a unit basis of the column space of $Z$.
4: For each $(i, t) \in [N] \times [T]$, $s^{(O)}(i, t) \leftarrow \|A_{iT - T + t}\|_2^2$.
5: For each pair $(i, t) \in [N] \times [T]$,
   $s(i, t) \leftarrow \min \left\{ 1, 2\lambda^{-1} \left( s^{(O)}(i, t) + \sum_{j=1}^{\min\{t-1, q\}} s^{(O)}(i, t - j) \right) \right\}$.
6: Pick a random sample $S \subseteq [N] \times [T]$ of $M$ pairs, where each $(i, t) \in S$ is selected with probability $\frac{s(i,t)}{\sum_{(i', t') \in [N] \times [T]} s(i', t')}$.
7: For each $(i, t) \in S$, $w(i, t) \leftarrow \frac{\sum_{(i', t') \in [N] \times [T]} s(i', t')}{M \cdot s(i,t)}$.
8: Output $(S, w)$.

---

## 4.2  Proof of Theorem 4.1

Algorithm 1 applies the FL framework (Feldman and Langberg [21]) that constructs coresets by importance sampling and the coreset size has been improved by [9]. The details of the unified FL framework can be found in Section B.2. The key is to verify the "pseudo-dimension" (Lemma 4.3) and "sensitivities" (Lemma 4.4) separately; summarized as follows.

**Upper bounding the pseudo-dimension.**  For preparation, we introduce a notion which measures the combinatorial complexity that plays the same role as VC-dimension [51].

**Definition 4.2 (Pseudo-dimension [21, 9])** *Given an arbitrary weight function $u : [N] \times [T] \to \mathbb{R}_{\geq 0}$, we define $\mathsf{range}_u(\zeta, r) = \left\{ (i, t) \in [N] \times [T] : u(i, t) \cdot \psi_{it}^{(G,q)}(\zeta) \leq r \right\}$ for every $\zeta \in \mathcal{P}_\lambda$ and*

$r \geq 0$. *The (pseudo-)dimension of GLSE is the largest integer $t$ such that there exists a weight function $u$ and a subset $A \subseteq \mathcal{X}$ of size $t$ satisfying that $|\{A \cap \mathsf{range}_u(\zeta, r) : \zeta \in \mathcal{P}_\lambda, r \geq 0\}| = 2^{|A|}$.*

We have the following lemma that upper bounds the pseudo-dimension of GLSE.

**Lemma 4.3 (Pseudo-dimension of GLSE)** *The pseudo-dimension $\dim$ is at most $O\left((q+d)qd\right)$.*

The proof can be found in Section D.1. The main idea is to apply the prior results [2, 53] which shows that the pseudo-dimension is polynomially dependent on the number of regression parameters ($q + d$ for GLSE) and the number of operations of individual regression objectives ($O(qd)$ for GLSE).

**Constructing a sensitivity function.** Next, we show that the function $s$ constructed in Line 5 of Algorithm 1 is indeed a sensitivity function of GLSE that measures the maximum influence for each $x_{it} \in X$; summarized by the following lemma.

**Lemma 4.4 (Total sensitivity of GLSE)** *Function $s : [N] \times [T] \to \mathbb{R}_{\geq 0}$ of Algorithm 1 satisfies that for any $(i,t) \in [N] \times [T]$, $s(i,t) \geq \sup_{\zeta \in \mathcal{P}} \frac{\psi_{it}^{(G,q)}(\zeta)}{\psi^{(G,q)}(\zeta)}$ and $\mathcal{G} := \sum_{(i,t) \in [N] \times [T]} s(i,t) = O(\lambda^{-1}qd)$. Moreover, the construction time of function $s$ is $O(NTq + NTd^2)$.*

The proof can be found in Section D.2. Intuitively, if the sensitivity $s(i,t)$ is large, e.g., close to 1, $\psi_{it}^{(G,q)}$ must contribute significantly to the objective with respect to some parameter $\zeta \in \mathcal{P}_\lambda$. The sampling ensures that we are likely to include such pair $(i,t)$ in the coreset for estimating $\psi(\zeta)$. Due to the fact that the objective function of GLSE is non-convex which is different from OLSE, bounding the sensitivity of each individual-time pair for GLSE becomes significantly difficult. To handle this difficulty, we develop a reduction of sensitivities from GLSE to OLSE (Line 5 of Algorithm 1), based on the relations between $\psi^{(G,q)}$ and $\psi^{(O)}$, i.e., for any $\zeta = (\beta, \rho) \in \mathcal{P}_\lambda$ we prove that $\psi_i^{(G,q)}(\zeta) \geq \lambda \cdot \psi_i^{(O)}(\beta)$ and $\psi_{it}^{(G,q)}(\zeta) \leq 2 \cdot \left(\psi_{it}^{(O)}(\beta) + \sum_{j=1}^{\min\{t-1,q\}} \psi_{i,t-j}^{(O)}(\beta)\right)$. The first inequality follows from the fact that the smallest eigenvalue of $\Omega_\rho^{-1}$ (the inverse covariance matrix induced by $\rho$) is at least $\lambda$. The intuition of the second inequality is from the form of function $\psi_{it}^{(G,q)}$, which relates to $\min\{t, q+1\}$ individual-time pairs, say $(x_{i,\min\{1,t-q\}}, y_{i,\min\{1,t-q\}}), \dots, (x_{it}, y_{it})$. Then it suffices to construct $s^{(O)}$ (Lines 2-4 of Algorithm 1), which reduces to the cross-sectional data setting and has total sensitivity at most $d + 1$ (Lemma D.3). Consequently, we conclude that the total sensitivity $\mathcal{G}$ of GLSE is $O(\lambda^{-1}qd)$ by the definition of $s$.

**Proof:** [Proof of Theorem 4.1] By Lemma 4.4, the total sensitivity $\mathcal{G}$ is $O(\lambda^{-1}qd)$. By Lemma 4.3, we let $\dim = O\left((q+d)qd\right)$. Pluging the values of $\mathcal{G}$ and $\dim$ in the FL framework [21, 9], we prove for the coreset size. For the running time, it costs $O(NTq + NTd^2)$ time to compute the sensitivity function $s$ by Lemma 4.4, and $O(NTd)$ time to construct an $\varepsilon$-coreset. This completes the proof. $\square$

## 5 Coresets for GLSE$_k$

Following from Section 4, we assume that the parameter space is $\mathcal{P}_\lambda^k = (\mathbb{R}^d \times B_{1-\lambda}^q)^k$ for some given constant $\lambda \in (0,1)$. Given a panel dataset $X \in \mathbb{R}^{N \times T \times d}$ and $Y \in \mathbb{R}^{N \times T}$, let $Z^{(i)} \in \mathbb{R}^{T \times (d+1)}$ denote a matrix whose $t$-th row is $(x_{it}, y_{it}) \in \mathbb{R}^{d+1}$ for all $t \in [T]$ ($i \in [N]$). Assume there exists constant $M \geq 1$ such that the input dataset satisfies the following property.

**Definition 5.1 ($M$-bounded dataset)** *Given $M \geq 1$, we say a panel dataset $X \in \mathbb{R}^{N \times T \times d}$ and $Y \in \mathbb{R}^{N \times T}$ is $M$-bounded if for any $i \in [N]$, the condition number of matrix $(Z^{(i)})^\top Z^{(i)}$ is at most $M$, i.e., $\max_{\beta \in \mathbb{R}^d} \frac{\psi_i^{(O)}(\beta)}{\|\beta\|_2^2 + 1} \leq M \cdot \min_{\beta \in \mathbb{R}^d} \frac{\psi_i^{(O)}(\beta)}{\|\beta\|_2^2 + 1}$.*

If there exists $i \in [N]$ and $\beta \in \mathbb{R}^d$ such that $\psi_i^{(O)}(\beta) = 0$, we let $M = \infty$. Specifically, if all $(Z^{(i)})^\top Z^{(i)}$ are identity matrix whose eigenvalues are all 1, i.e., for any $\beta$, $\psi_i^{(O)}(\beta) = \|\beta\|_2^2 + 1$, we can set $M = 1$. Another example is that if $n \gg d$ and all elements of $Z^{(i)}$ are independently and identically distributed standard normal random variables, then the condition number of matrix $(Z^{(i)})^\top Z^{(i)}$

is upper bounded by some constant with high probability (and constant in expectation) [10, 46], which may also imply $M = O(1)$. The main theorem is as follows.

**Theorem 5.2 (Coresets for GLSE$_k$)** *There exists a randomized algorithm that given an $M$-bounded ($M \geq 1$) panel dataset $X \in \mathbb{R}^{N \times T \times d}$ and $Y \in \mathbb{R}^{N \times T}$, constant $\varepsilon, \lambda \in (0,1)$ and integers $q, k \geq 1$, with probability at least 0.9, constructs an $\varepsilon$-coreset for GLSE$_k$ of size $O\left(\varepsilon^{-4}\lambda^{-2}Mk^2 \max\left\{q^7d^4, q^5d^6\right\} \cdot \log \frac{Mq}{\lambda} \log \frac{Mkd}{\lambda}\right)$ and runs in time $O(NTq + NTd^2)$.*

Similar to GLSE, this coreset for GLSE$_k$ ($k \geq 2$) contains at most $(q+1) \cdot O\left(\varepsilon^{-4}\lambda^{-2}Mk^2 \max\left\{q^7d^4, q^5d^6\right\} \cdot \log \frac{Mq}{\lambda} \log \frac{kd}{\lambda}\right)$ points $(x_{it}, y_{it})$, which is independent of both $N$ and $T$ when $M$ is constant. Note that the size contains an addtional factor $M$ which can be unbounded. Our algorithm is summarized in Algorithm 2 and the proof of Theorem 5.2 can be found in Section E. Due to the space limit, we outline Algorithm 2 and discuss the novelty in the following.

---

**Algorithm 2:** CGLSE$_k$: Coreset construction of GLSE$_k$

---

**Input:** an $M$-bounded (constant $M \geq 1$) panel dataset $X \in \mathbb{R}^{N \times T \times d}$ and $Y \in \mathbb{R}^{N \times T}$, constant $\varepsilon, \lambda \in (0, 1)$, integers $k, q \geq 1$ and parameter space $\mathcal{P}_\lambda^k$.
**Output:** a subset $S \subseteq [N] \times [T]$ together with a weight function $w : S \to \mathbb{R}_{\geq 0}$.

---

   % Constructing a subset of individuals
1: $\Gamma \leftarrow O\left(\varepsilon^{-2}\lambda^{-1}Mk^2 \max\left\{q^4d^2, q^3d^3\right\} \cdot \log \frac{Mq}{\lambda}\right)$.
2: For each $i \in [N]$, let matrix $Z^{(i)} \in \mathbb{R}^{T \times (d+1)}$ be whose $t$-th row is $z_t^{(i)} = (x_{it}, y_{it}) \in \mathbb{R}^{d+1}$.
3: For each $i \in [N]$, construct the SVD decomposition of $Z^{(i)}$ and compute

$$u_i := \lambda_{\max}((Z^{(i)})^\top Z^{(i)}) \text{ and } \ell_i := \lambda_{\min}((Z^{(i)})^\top Z^{(i)}).$$

4: For each $i \in [N]$, $s^{(O)}(i) \leftarrow \frac{u_i}{u_i + \sum_{i' \neq i} \ell_{i'}}$.
5: For each $i \in [N]$, $s(i) \leftarrow \min\left\{1, \frac{2(q+1)}{\lambda} \cdot s^{(O)}(i)\right\}$.
6: Pick a random sample $I_S \subseteq [N]$ of size $M$, where each $i \in I_S$ is selected with probability $\frac{s(i)}{\sum_{i' \in [N]} s(i')}$.
7: For each $i \in I_S$, $w'(i) \leftarrow \frac{\sum_{i' \in [N]} s(i')}{\Gamma \cdot s(i)}$.

---

   % Constructing a subset of time periods for each selected individual
8: For each $i \in I_S$, apply CGLSE($X_i, y_i, \frac{\varepsilon}{3}, \frac{1}{20\Gamma}, \lambda, q$) and construct a subset $J_{S,i} \subseteq [T]$ together with a weight function $w^{(i)} : J_{S,i} \to \mathbb{R}_{\geq 0}$.
9: Let $S \leftarrow \{(i, t) \in [N] \times [T] : i \in I_S, t \in J_{S,i}\}$.
10: For each $(i, t) \in S$, $w(i, t) \leftarrow w'(i) \cdot w^{(i)}(t)$.
11: Output $(S, w)$.

---

**Remark 5.3** *Algorithm 2 is a two-staged framework, which captures the* min *operations in GLSE$_k$.*

**First stage.** *We construct an $\frac{\varepsilon}{3}$-coreset $I_S \subseteq [N]$ together with a weight function $w' : I_S \to \mathbb{R}_{\geq 0}$ satisfying $\sum_{i \in I_S} w'(i) \cdot \psi_i^{(G,q,k)}(\zeta) \in (1 \pm \varepsilon) \cdot \psi^{(G,q,k)}(\zeta)$. The idea is similar to Algorithm 1 except that we consider $N$ sub-functions $\psi_i^{(G,q,k)}$ instead of $NT$. In Lines 2-4 of Algorithm 2, we first construct a sensitivity function $s^{(O)}$ of OLSE$_k$. The definition of $s^{(O)}$ captures the impact of* min *operations in the objective function of OLSE$_k$ and the total sensitivity of $s^{(O)}$ is guaranteed to be upper bounded by Definition 5.1. The key is showing that the maximum influence of individual $i$ is at most $\frac{u_i}{u_i + \sum_{j \neq i} \ell_j}$ (Lemma E.3), which implies that the total sensitivity of $s^{(O)}$ is at most $M$. Then in Line 5, we construct a sensitivity function $s$ of GLSE$_k$, based on a reduction from $s^{(O)}$ (Lemma E.4). The key observations are that for any $\zeta = (\beta, \rho) \in \mathcal{P}_\lambda$ we have $\psi_i^{(G,q)}(\zeta) \geq \lambda \cdot \psi_i^{(O)}(\beta)$*

*that provides an upper bound of the individual objective gap between GLSE and OLSE, and for any $\zeta = (\beta^{(1)}, \ldots, \beta^{(k)}, \rho^{(1)}, \ldots, \rho^{(k)}) \in \mathcal{P}^k$, $\psi_i^{(G,q,k)}(\zeta) \leq 2(q+1) \cdot \min_{l \in [k]} \psi_i^{(O)}(\beta^{(l)})$, that provides a lower bound of the individual objective gap between $\mathrm{GLSE}_k$ and $\mathrm{OLSE}_k$.*

**Second stage.** In Line 8, for each $i \in I_S$, apply $\mathsf{CGLSE}(X_i, y_i, \frac{\varepsilon}{3}, \frac{1}{20 \cdot |I_S|}, \lambda, q)$ and construct a subset $J_{S,i} \subseteq [T]$ together with a weight function $w^{(i)} : J_{S,i} \to \mathbb{R}_{\geq 0}$. Output $S = \{(i,t) \in [N] \times [T] : i \in I_S, t \in J_{S,i}\}$ together with a weight function $w : S \to \mathbb{R}_{\geq 0}$ defined as follows: for any $(i,t) \in S$, $w(i,t) := w'(i) \cdot w^{(i)}(t)$.

We also provide a lower bound theorem which shows that the size of a coreset for $\mathrm{GLSE}_k$ can be up to $\Omega(N)$. It indicates that the coreset size should contain additional factors than $k, q, d, 1/\varepsilon$, which reflects the reasonability of the $M$-bounded assumption. The proof can be found in Section E.

**Theorem 5.4 (Size lower bound of $\mathrm{GLSE}_k$)** *Let $T = 1$ and $d = k = 2$ and $\lambda \in (0,1)$. There exists $X \in \mathbb{R}^{N \times T \times d}$ and $Y \in \mathbb{R}^{N \times T}$ such that any 0.5-coreset for $\mathrm{GLSE}_k$ should have size $\Omega(N)$.*

# 6 Empirical results

We implement our coreset algorithms for GLSE, and compare the performance with uniform sampling on synthetic datasets and a real-world dataset. The experiments are conducted by PyCharm on a 4-Core desktop CPU with 8GB RAM.[1]

**Datasets.** We experiment using **synthetic** datasets with $N = T = 500$ ($250k$ observations), $d = 10$, $q = 1$ and $\lambda = 0.2$. For each individual $i \in [N]$, we first generate a mean vector $\overline{x}_i \in \mathbb{R}^d$ by first uniformly sampling a unit vector $x'_i \in \mathbb{R}^d$, and a length $\tau \in [0,5]$, and then letting $\overline{x}_i = \tau x'_i$. Then for each time period $t \in [T]$, we generate observation $x_{it}$ from a multivariate normal distribution $N(\overline{x}_i, \|\overline{x}_i\|_2^2 \cdot I)$ [50].[2] Next, we generate outcomes $Y$. First, we generate a regression vector $\beta \in \mathbb{R}^d$ from distribution $N(0, I)$. Then we generate an autoregression vector $\rho \in \mathbb{R}^q$ by first uniformly sampling a unit vector $\rho' \in \mathbb{R}^q$ and a length $\tau \in [0, 1-\lambda]$, and then letting $\rho = \tau \rho'$. Based on $\rho$, we generate error terms $e_{it}$ as in Equation (2). To assess performance robustness in the presence of outliers, we simulate another dataset replacing $N(0, I)$ in Equation (2) with the heavy tailed **Cauchy**(0,2) distribution [38]. Finally, the outcome $y_{it} = x_{it}^\top \beta + e_{it}$ is the same as Equation (1).

We also experiment on a **real-world** dataset involving the prediction of monthly profits from customers for a credit card issuer as a function of demographics, past behaviors, and current balances and fees. The panel dataset consisted of 250k observations: 50 months of data ($T = 50$) from 5000 customers ($N = 5000$) with 11 features ($d = 11$). We set $q = 1$ and $\lambda = 0.2$.

**Baseline and metrics.** As a baseline coreset, we use uniform sampling (**Uni**), perhaps the simplest approach to construct coresets: Given an integer $\Gamma$, uniformly sample $\Gamma$ individual-time pairs $(i,t) \in [N] \times [T]$ with weight $\frac{NT}{\Gamma}$ for each.

Given regression parameters $\zeta$ and a subset $S \subseteq [N] \times [T]$, we define the *empirical error* as $\left| \frac{\psi_S^{(G,q)}(\zeta)}{\psi^{(G,q)}(\zeta)} - 1 \right|$. We summarize the empirical errors $e_1, \ldots, e_n$ by maximum, average, standard deviation (std) and root mean square error (RMSE), where RMSE$= \sqrt{\frac{1}{n} \sum_{i \in [n]} e_i^2}$. By penalizing larger errors, RMSE combines information in both average and standard deviation as a performance metric,. The running time for solving GLSE on dataset $X$ and our coreset $S$ are $T_X$ and $T_S$ respectively. $T_C$ is the running time for coreset $S$ construction .

**Simulation setup.** We vary $\varepsilon = 0.1, 0.2, 0.3, 0.4, 0.5$ and generate 100 independent random tuples $\zeta = (\beta, \rho) \in \mathbb{R}^{d+q}$ (the same as described in the generation of the synthetic dataset). For each $\varepsilon$, we run our algorithm $\mathsf{CGLSE}$ and **Uni** to generate coresets. We guarantee that the total number of sampled individual-time pairs of $\mathsf{CGLSE}$ and **Uni** are the same. We also implement IRLS [32] for solving GLSE. We run IRLS on both the full dataset and coresets and record the runtime.

Table 1: performance of $\varepsilon$-coresets for GLSE w.r.t. varying $\varepsilon$. We report the maximum/average/standard deviation/RMSE of the empirical error w.r.t. the 100 tuples of generated regression parameters for our algorithm CGLSE and **Uni**. Size is the # of sampled individual-time pairs, for both CGLSE and **Uni**. $T_C$ is construction time (seconds) of our coresets. $T_S$ and $T_X$ are the computation time (seconds) for GLSE over coresets and the full dataset respectively. "Synthetic (G)" and "Synthetic (C)" represent synthetic datasets with Gaussian errors and Cauchy errors respectively.

| | $\varepsilon$ | max. emp. err. | | avg./std./RMSE of emp. err. | | size | $T_C$ | $T_C + T_S$ | $T_X$ (s) |
| | | CGLSE | Uni | CGLSE | Uni | | | | |
|---|---|---|---|---|---|---|---|---|---|
| synthetic (G) | 0.1 | **.005** | .015 | .001/.001/.002 | .007/.004/.008 | 116481 | 2 | 372 | 458 |
| | 0.2 | **.018** | .029 | .006/.004/.008 | .010/.007/.013 | 23043 | 2 | 80 | 458 |
| | 0.3 | **.036** | .041 | .011/.008/.014 | .014/.010/.017 | 7217 | 2 | 29 | 458 |
| | 0.4 | **.055** | .086 | .016/.012/.021 | .026/.020/.032 | 3095 | 2 | 18 | 458 |
| | 0.5 | **.064** | .130 | .019/.015/.024 | .068/.032/.075 | 1590 | 2 | 9 | 458 |
| synthetic (C) | 0.1 | **.001** | .793 | .000/.000/.001 | .744/.029/.745 | 106385 | 2 | 1716 | 4430 |
| | 0.2 | **.018** | .939 | .013/.003/.014 | .927/.007/.927 | 21047 | 2 | 346 | 4430 |
| | 0.3 | **.102** | .937 | .072/.021/.075 | .860/.055/.862 | 6597 | 2 | 169 | 4430 |
| | 0.4 | **.070** | .962 | .051/.011/.053 | .961/.001/.961 | 2851 | 2 | 54 | 4430 |
| | 0.5 | **.096** | .998 | .060/.026/.065 | .992/.004/.992 | 472 | 2 | 41 | 4430 |
| real-world | 0.1 | **.029** | .162 | .005/.008/.009 | .016/.026/.031 | 50777 | 3 | 383 | 2488 |
| | 0.2 | **.054** | .154 | .017/.004/.017 | .012/.024/.026 | 13062 | 3 | 85 | 2488 |
| | 0.3 | **.187** | .698 | .039/.038/.054 | .052/.106/.118 | 5393 | 3 | 24 | 2488 |
| | 0.4 | **.220** | .438 | .019/.033/.038 | .050/.081/.095 | 2734 | 3 | 20 | 2488 |
| | 0.5 | **.294** | 1.107 | .075/.038/.084 | .074/.017/.183 | 1534 | 3 | 16 | 2488 |

**Results.** Table 1 summarizes the accuracy-size trade-off of our coresets for GLSE for different error guarantees $\varepsilon$. The maximum empirical error of **Uni** is always larger than that of our coresets (1.16-793x). Further, there is no error guarantee with **Uni**, but errors are always below the error guarantee with our coresets. The speed-up with our coresets relative to full data ($\frac{T_X}{T_C + T_S}$) in solving GLSE is 1.2x-108x. To achieve the maximum empirical error of .294 for GLSE in the real-world data, only 1534 individual-time pairs (0.6%) are necessary for CGLSE. With **Uni**, to get the closest maximum empirical error of 0.438, at least 2734 individual-time pairs) (1.1%) is needed; i.e.., CGLSE achieves a smaller empirical error with a smaller sized coreset. Though **Uni** may sometimes provide lower average error than CGLSE, it *always* has higher RMSE, say 1.2-745x of CGLSE. When there are outliers as with Cauchy, our coresets perform even better on all metrics relative to **Uni**. This is because CGLSE captures tails/outliers in the coreset, while **Uni** does not. Figure 1 in Section F presents the boxplots of the empirical errors.

## 7   Conclusion, limitations, and future work

This paper initiates a theoretical study of coreset construction for regression problems with panel data. We formulate the definitions of coresets for several variants of $\ell_2$-regression, including OLSE, GLSE, and GLSE$_k$. For each variant, we propose efficient algorithms that construct a coreset of size independent of both $N$ and $T$, based on the FL framework. Our empirical results indicate that our algorithms can accelerate the evaluation time and perform significantly better than uniform sampling.

For GLSE$_k$, our coreset size contains a factor $M$, which may be unbounded and result in a coreset of size $\Omega(N)$ in the worst case. In practice, if $M$ is large, each sensitivity $s(i)$ in Line 5 of Algorithm 2 will be close or even equal to 1. In this case, $I_S$ is drawn from all individuals via uniform sampling which weakens the performance of Algorithm 2 relative to **Uni**. Future research should investigate whether a different assumption than the $M$-bound can generate a coreset of a smaller size.

There are several directions for future work. Currently, $q$ and $d$ have a relatively large impact on coreset size; future work needs to reduce this effect. This will advance the use of coresets for machine learning, where $d$ is typically large, and $q$ is large in high frequency data. This paper focused on coreset construction for panel data with $\ell_2$-regression. The natural next steps would be to construct coresets with panel data for other regression problems, e.g., $\ell_1$-regression, generalized linear models and logistic regression, and beyond regression to other supervised machine learning algorithms.

## Broader impact

Many organizations have to routinely outsource data processing to external consultants and statisticians. A major practical challenge for organizations in doing this is to minimize issues of data security in terms of exposure of their data for potential abuse. Further, minimization of such exposure is considered as necessary due diligence by laws such as GDPR and CCPA which mandates firms to minimize security breaches that violate the privacy rights of the data owner [45, 34]. Coreset based approaches to sharing data for processing can be very valuable for firms in addressing data security and to be in compliance with privacy regulations like GDPR and CCPA.

Obtaining unbiased estimates of the regression relationships from observational data is often very critical for making correct policy decisions in economics and many social sciences. Panel data is one critical ingredient for obtaining unbiased estimates. As ML methods are being adopted by many social scientists [5], ML scholars are becoming sensitive to these issues and our work in using coreset methods for panel data can have significant impact for these scholars.

We don't foresee immediate negative impact from using our method. However, one concern might be that coresets constructed and shared for one purpose or model may be used by the data processor for other kinds of models, which may lead to erroneous conclusions. There is also the potential for issues of fairness to arise as different groups may not be adequately represented in the coreset without incorporating fairness constraints [29]. These issues may need to be explored in future research.

## Acknowledgements

This research was conducted when LH was at Yale and was supported in part by an NSF CCF-1908347 grant.

## Footnotes

[1]Codes are in `https://github.com/huanglx12/Coresets-for-regressions-with-panel-data`.

[2]The assumption that the covariance of each individual is proportional to $\|\overline{x}_i\|_2^2$ is common in econometrics. We also fix the last coordinate of $x_{it}$ to be 1 to capture individual specific fixed effects.

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
