[Supplementary Material]

# A    Discussion of the generative model (1)

In this section, we discuss the equivalence between the generative model (1) and the random effects estimator. In random effects estimators, there exist additional individual specified effects $\alpha_i \in \mathbb{R}$, i.e.,

$$y_{it} = x_{it}^\top \beta_i + \alpha_i + e_{it}, \tag{3}$$

and we assume that all individual effects are drawn from a normal distribution, i.e.,

$$\alpha_i \sim N(\mu, \sigma_0^2), \qquad \forall i \in [N].$$

where $\mu \in \mathbb{R}$ is the mean and $\sigma_0^2 \in \mathbb{R}_{\geq 0}$ is the covariance of an unknown normal distribution. By Equation (3), for any $i \in [N]$, we let $\alpha_i = \mu + \varepsilon_i$ where $\varepsilon_i \sim N(0, \sigma_0^2)$. Then Equation (3) can be rewritten as

$$y_{it} = x_{it}^\top \beta_i + \mu + (\varepsilon_i + e_{it}).$$

Let $\Omega \in \mathbb{R}^{T \times T}$ denote the covariance matrix among error terms $e_{it}$. Next, we simplify $\varepsilon_i + e_{it}$ by $e'_{it}$. Consequently, error terms $e'_{it}$ satisfy that

$$
\begin{aligned}
&\mathsf{Exp}[e'_{it}] = 0, && \forall (i,t) \in [N] \times [T]; \\
&\mathsf{Cov}(e'_{it}, e'_{i't'}) = 0 && \forall i \neq i' \\
&\mathsf{Cov}(e'_{it}, e'_{it'}) = \Omega_{tt'} + \sigma_0^2 = \Omega'_{tt'} && \forall i \in [N], t, t' \in [T].
\end{aligned}
$$

By this assumption, a random effects estimator can be defined by the following:

$$\min_{\beta, \Omega} \sum_{i \in [N]} (y_i - X_i \beta_i - \mu \cdot \mathbf{1})^\top (\Omega')^{-1} (y_i - X_i \beta_i - \mu \cdot \mathbf{1}).$$

Thus, we verify that the random effects estimator is equivalent to the generative model (1).

# B    Revisit the coreset definition and the Feldman-Langberg framework

In this section, we derive the coreset definitions in Section 3 from coreset of a query space [21, 52]. Then we revisit the Feldman-Langberg framework using the language of query spaces, which helps us to complete the missing proofs in Sections 4 and 5.

## B.1    Revisit the coreset definition

**OLSE and GLSE.**    We first introduce a generalized definition: coresets of a query space, which captures the coreset definition for OLSE and GLSE.

**Definition B.1 (Query space [21, 9])** *Let $\mathcal{X}$ be a index set together with a weight function $u : \mathcal{X} \to \mathbb{R}_{\geq 0}$. Let $\mathcal{P}$ be a set called queries, and $\psi_x : \mathcal{P} \to \mathbb{R}_{\geq 0}$ be a given loss function w.r.t. $x \in \mathcal{X}$. The total cost of $\mathcal{X}$ with respect to a query $\zeta \in \mathcal{P}$ is $\psi(\zeta) := \sum_{x \in \mathcal{X}} u(x) \cdot \psi_x(\zeta)$. The tuple $(\mathcal{X}, u, \mathcal{P}, \psi)$ is called a query space. Specifically, if $u(x) = 1$ for all $x \in \mathcal{X}$, we use $(\mathcal{X}, \mathcal{P}, \psi)$ for simplicity.*

Intuitively, $\psi$ represents a linear combination of weighted functions indexed by $\mathcal{X}$, and $\mathcal{P}$ represents the ground set of $\psi$. Due to the separability of $\psi$, we have the following coreset definition.

**Definition B.2 (Coresets of a query space [21, 9])** *Let $(\mathcal{X}, u, \mathcal{P}, \psi)$ be a query space and $\varepsilon \in (0, 1)$ be an error parameter. An $\varepsilon$-coreset of $(\mathcal{X}, u, \mathcal{P}, \psi)$ is a weighted set $S \subseteq \mathcal{X}$ together with a weight function $w : S \to \mathbb{R}_{\geq 0}$ such that for any $\zeta \in \mathcal{P}$, $\psi_S(\zeta) := \sum_{x \in S} w(x) \cdot \psi_x(\zeta) \in (1 \pm \varepsilon) \cdot \psi(\zeta)$.*

Here, $\psi_S$ is a computation function over the coreset that is used to estimate the total cost of $\mathcal{X}$. By Definitions 2.2 and 2.3, the regression objectives of OLSE and GLSE can be decomposed into $NT$ sub-functions. Thus, we can apply the above definition to define coresets for OLSE and GLSE. Note that OLSE is a special case of GLSE for $q = 0$. Thus, we only need to provide the coreset definition for GLSE. We let $u = 1$ and $\mathcal{P} = \mathbb{R}^d \times B^q$. The index set of GLSE has the following form:

$$Z^{(G,q)} = \left\{ z_{it} = \left( x_{i, \max\{1, t-q\}}, y_{i, \max\{1, t-q\}}, \ldots x_{it}, y_{it} \right) : (i, t) \in [N] \times [T] \right\},$$

where each element $z_{it}$ consists of at most $q + 1$ observations. Also, for every $z_{it} \in Z^{(G,q)}$ and $\zeta = (\beta, \rho) \in \mathcal{P}$, the cost function $\psi_{it}$ is: if $t = 1$, $\psi_{it}^{(G,q)}(\zeta) = (1 - \|\rho\|_2^2) \cdot (y_{i1} - x_{i1}^\top \beta)^2$; and if $t \neq 1$, $\psi_{it}^{(G,q)}(\zeta) = \left( (y_{it} - x_{it}^\top \beta) - \sum_{j=1}^{\min\{t-1,q\}} \rho_j (y_{i,t-j} - x_{i,t-j}^\top \beta) \right)^2$. Thus, $(Z^{(G,q)}, \mathcal{P}, \psi^{(G,q)})$ is a query space of GLSE.[3] We conclude that the weighted set $S$ in Definition 3.1 is exactly an $\varepsilon$-coreset of the query space $(Z^{(G,q)}, \mathcal{P}, \psi^{(G,q)})$. Specifically, for OLSE, the parameter space is $\mathbb{R}^d$ since $q = 0$, and the corresponding index set is $Z^{(O)} = \{z_{it} = (x_{it}, y_{it}) : (i,t) \in [N] \times [T]\}$. Consequently, the query space of OLSE is $(Z^{(O)}, \mathbb{R}^d, \psi^{(O)})$.

**GLSE$_k$.** Let $u = 1$, $\mathcal{P}^k = \left(\mathbb{R}^d \times B^q\right)^k$, and $Z^{(G,q,k)} = \{z_i = (x_{i1}, y_{i1}, \ldots, x_{iT}, y_{iT}) : i \in [N]\}$. We can regard $(Z^{(G,q,k)}, \mathcal{P}^k, \psi^{(G,q,k)})$ as a query space of GLSE$_k$. By Definition B.2, an $\varepsilon$-coreset of $(Z^{(G,q,k)}, \mathcal{P}^k, \psi^{(G,q,k)})$ is a subset $I_S \subseteq [N]$ together with a weight function $w' : I_S \to \mathbb{R}_{\geq 0}$ such that for any $\zeta \in \mathcal{P}^k$, Inequality (**??**) holds, i.e.,

$$\sum_{i \in I_S} w'(i) \cdot \psi_i^{(G,q,k)}(\zeta) \in (1 \pm \varepsilon) \cdot \psi^{(G,q,k)}(\zeta).$$

However, each $z_i \in Z^{(G,q,k)}$ consists of $T$ observations, and hence, the number of points in this coreset $S$ is $T \cdot |S|$. To avoid the size dependence of $T$, we propose Definition 3.2 for GLSE$_k$. The intuition is to further select a subset of time periods to estimate $\psi_i^{(G,q,k)}$.

## B.2 Revisit the Feldman-Langberg framework

We also revisit the FL framework stated in Section **??**, which is specific for GLSE. We state the general FL framework that designs for a query space, which captures the query spaces for GLSE and GLSE$_k$ in the last subsection. For preparation, we first give the notion of sensitivity which measures the maximum influence for each point $x \in \mathcal{X}$.

**Definition B.3 (Sensitivity [21, 9])** *Given a query space $(\mathcal{X}, u, \mathcal{P}, \psi)$, the sensitivity of a point $x \in \mathcal{X}$ is $s(x) := \sup_{\zeta \in \mathcal{P}} \frac{u(x) \cdot \psi_x(\zeta)}{\psi(\mathcal{X}, u, \zeta)}$. The total sensitivity of the query space is $\sum_{x \in \mathcal{X}} s(x)$.*

Intuitively, if the sensitivity $s(x)$ of some point $x$ is large, e.g., close to 1, $x$ must contribute significantly to the objective with respect to some query $\zeta \in \mathcal{P}$. The sampling ensures that we are likely to include such an $x$ in the coreset for estimating $\psi(\zeta)$. We also rewrite Definition 4.2 in the language of a query space.

**Definition B.4 (Pseudo-dimension [21, 9])** *For a query space $(\mathcal{X}, u, \mathcal{P}, \psi)$, we define $\mathsf{range}(\zeta, r) = \{x \in \mathcal{X} : u(x) \cdot \psi_x(\zeta) \leq r\}$ for every $\zeta \in \mathcal{P}$ and $r \geq 0$. The (pseudo-)dimension of $(\mathcal{X}, u, \mathcal{P}, \psi)$ is the largest integer $t$ such that there exists a subset $A \subseteq \mathcal{X}$ of size $t$ satisfying that $|\{A \cap \mathsf{range}(\zeta, r) : \zeta \in \mathcal{P}, r \geq 0\}| = 2^{|A|}$.*

Pseudo-dimension plays the same role as VC-dimension [51]. Specifically, if the range of $\psi$ is $\{0, 1\}$ and $u = 1$, pseudo-dimension can be regarded as a generalization of VC-dimension to function spaces. Now we are ready to describe the FL framework in the language of a query space.

**Theorem B.5 (FL framework [21, 9])** *Let $(\mathcal{X}, u, \mathcal{P}, \psi)$ be a given query space and $\varepsilon, \delta \in (0, 1)$. Let $\dim$ be an upper bound of the pseudo-dimension of every query space $(\mathcal{X}, u', \mathcal{P}, \psi)$ over $u'$. Suppose $s : \mathcal{X} \to \mathbb{R}_{\geq 0}$ is a function satisfying that for any $x \in \mathcal{X}$, $s(x) \geq \sup_{\zeta \in \mathcal{P}} \frac{u(x) \cdot \psi_x(\zeta)}{\psi(\mathcal{X}, u, \zeta)}$, and $\mathcal{G} := \sum_{x \in \mathcal{X}} s(x)$. Let $S \subseteq \mathcal{X}$ be constructed by taking $O\left(\varepsilon^{-2}\mathcal{G}(\dim \cdot \log \mathcal{G} + \log(1/\delta))\right)$ samples, where each sample $x \in \mathcal{X}$ is selected with probability $\frac{s(x)}{\mathcal{G}}$ and has weight $w(x) := \frac{\mathcal{G}}{|S| \cdot s(x)}$. Then, with probability at least $1 - \delta$, $S$ is an $\varepsilon$-coreset of $(\mathcal{X}, u, \mathcal{P}, \psi)$.*

# C Existing results and approaches for OLSE

We note that finding an $\varepsilon$-coreset of $X$ for OLSE can be reduced to finding an $\varepsilon$-coreset for least-squares regression with cross-sectional data. For completeness, we summarize the following theorems for OLSE whose proofs mainly follow from the literature.

**Theorem C.1** ($\varepsilon$-**Coresets for OLSE [8]**) *There exists a deterministic algorithm that for any given observation matrix $X \in \mathbb{R}^{N \times T \times d}$, outcome matrix $Y \in \mathbb{R}^{N \times T}$, a collection $\mathcal{B} \subseteq \mathbb{R}^d$ and constant $\varepsilon \in (0,1)$, constructs an $\varepsilon$-coreset of size $O(d/\varepsilon^2)$ of OLSE, with running time $T_{SVD} + O(NTd^3/\varepsilon^2)$ where $T_{SVD}$ is the time needed to compute the left singular vectors of a matrix in $\mathbb{R}^{NT \times (d+1)}$.*

**Theorem C.2 (Accurate coresets for OLSE [33])** *There exists a deterministic algorithm that for any given observation matrix $X \in \mathbb{R}^{N \times T \times d}$, outcome matrix $Y \in \mathbb{R}^{N \times T}$, a collection $\mathcal{B} \subseteq \mathbb{R}^d$, constructs an accurate coreset of size $O(d^2)$ of OLSE, with running time $O(NTd^2 + d^8 \log(NT/d))$.*

## C.1 Proof of Theorem C.1

We first prove Theorem C.1 and propose the corresponding algorithm that constructs an $\varepsilon$-coreset. Recall that $\mathcal{B} \subseteq \mathbb{R}^d$ denotes the domain of possible vectors $\beta$.

**Proof:** [Proof of Theorem C.1] Construct a matrix $A \in \mathbb{R}^{NT \times d}$ by letting the $(iT - T + t)$-th row of $A$ be $x_{it}$ for $(i,t) \in [N] \times [T]$. Similarly, construct a vector $\mathbf{b} \in \mathbb{R}^{NT}$ by letting $\mathbf{b}_{iT-T+t} = y_{it}$. Then for any $\beta \in \mathcal{B}$, we have

$$\psi^{(O)}(\beta) = \|A\beta - \mathbf{b}\|_2^2.$$

Thus, finding an $\varepsilon$-coreset of $X$ of OLSE is equivalent to finding a row-sampling matrix $S \in \mathbb{R}^{m \times NT}$ whose rows are basis vectors $e_{i_1}^\top, \ldots, e_{i_m}^\top$ and a rescaling matrix $W \in \mathbb{R}_{\geq 0}^{m \times m}$ that is a diagonal matrix such that for any $\beta \in \mathcal{B}$,

$$\|WS(A\beta - \mathbf{b})\|_2^2 \in (1 \pm \varepsilon) \cdot \|A\beta - \mathbf{b}\|_2^2.$$

By Theorem 1 of [8], we only need $m = O(d/\varepsilon^2)$ which completes the proof of correctness. Note that Theorem 1 of [8] only provides a theoretical guarantee of a weak-coreset which only approximately preserves the optimal least-squares value. However, by the proof of Theorem 1 of [8], their coreset indeed holds for any $\beta \in \mathbb{R}^d$.

The running time also follows from Theorem 1 of [8], which can be directly obtained by the algorithm stated below. $\qquad\square$

**Algorithm in [8].** We then introduce the approach of [8] as follows. Suppose we have inputs $A \in \mathbb{R}^{n \times d}$ and $\mathbf{b} \in \mathbb{R}^n$.

1. Compute the SVD of $Y = [A, b] \in \mathbb{R}^{n \times (d+1)}$. Let $Y = U\Sigma V^\top$ where $U \in \mathbb{R}^{n \times (d+1)}, \Sigma \in \mathbb{R}^{(d+1) \times (d+1)}$ and $V \in \mathbb{R}^{(d+1) \times (d+1)}$.

2. By Lemma 2 of [8] which is based on Theorem 3.1 of [7], we deterministically construct sampling and rescaling matrices $S \in \mathbb{R}^{m \times n}$ and $W \in \mathbb{R}^{m \times m}$ ($m = O(d/\varepsilon^2)$) such that for any $y \in \mathbb{R}^{d+1}$,

$$\|WSUy\|_2^2 \in (1 \pm \varepsilon) \cdot \|Uy\|_2^2.$$

   The construction time is $O(nd^3/\varepsilon^2)$.

3. Output $S$ and $W$.

## C.2 Proof of Theorem C.2

Next, we prove Theorem C.2 and propose the corresponding algorithm that constructs an accurate coreset.

**Proof:** [Proof of Theorem C.2] The proof idea is similar to that of Theorem C.1. Again, we construct a matrix $A \in \mathbb{R}^{NT \times d}$ by letting the $(iT - T + t)$-th row of $A$ be $x_{it}$ for $(i, t) \in [N] \times [T]$. Similarly, construct a vector $\mathbf{b} \in \mathbb{R}^{NT}$ by letting $\mathbf{b}_{iT-T+t} = y_{it}$. Then for any $\beta \in \mathcal{B}$, we have

$$\psi^{(O)}(\beta) = \|A\beta - \mathbf{b}\|_2^2.$$

Thus, finding an $\varepsilon$-coreset of $X$ of OLSE is equivalent to finding a row-sampling matrix $S \in \mathbb{R}^{m \times NT}$ whose rows are basis vectors $e_{i_1}^\top, \ldots, e_{i_m}^\top$ and a rescaling matrix $W \in \mathbb{R}_{\geq 0}^{m \times m}$ that is a diagonal matrix such that for any $\beta \in \mathcal{B}$,

$$\|WS(A\beta - \mathbf{b})\|_2^2 = \|A\beta - \mathbf{b}\|_2^2.$$

By Theorem 3.2 of [33], we only need $m = (d+1)^2 + 1 = O(d^2)$. Moreover, we can construct matrices $W$ and $S$ in $O(NTd^2 + d^8 \log(NT/d))$ time by applying $n = NT$, and $k = 2(d+1)$ in Theorem 3.2 of [33]. $\qquad\square$

**Main approach in [33].** Suppose we have inputs $A \in \mathbb{R}^{n \times d}$ and $\mathbf{b} \in \mathbb{R}^n$. Let $A' = [A, \mathbf{b}] \in \mathbb{R}^{n \times (d+1)}$ For any $\beta \in \mathbb{R}^d$, we let $\beta' = (\beta, -1) \in \mathbb{R}^{d+1}$ and have that

$$\|A\beta - \mathbf{b}\|_2^2 = \|A'\beta'\|_2^2 = (\beta')^\top (A')^\top A'\beta'.$$

The main idea of [33] is to construct a sub-matrix $C \in \mathbb{R}^{((d+1)^2+1) \times (d+1)}$ of $A'$ whose rows are of the form $w_i \cdot (a_i, \mathbf{b}_i)^\top$ for some $i \in [n]$ and $w_i \geq 0$, such that $C^\top C = (A')^\top A'$. Then we have for any $\beta \in \mathbb{R}^d$,

$$\|C\beta'\|_2^2 = (\beta')^\top C^\top C\beta' = (\beta')^\top (A')^\top A'\beta' = \|A\beta - \mathbf{b}\|_2^2.$$

By the definition of $C$, there exists a row-sampling matrix $S$ and a rescaling matrix $W$ such that $C = WSA'$.

We then discuss why such a sub-matrix $C$ exists. The main observation is that $(A')^\top A' \in \mathbb{R}^{(d+1) \times (d+1)}$ and

$$(A')^\top A' = \sum_{i \in [n]} (a_i, \mathbf{b}_i) \cdot (a_i, \mathbf{b}_i)^\top.$$

Thus, $\frac{1}{n} \cdot (A')^\top A'$ is inside the convex hull of $n$ matrices $(a_i, \mathbf{b}_i) \cdot (a_i, \mathbf{b}_i)^\top \in \mathbb{R}^{(d+1) \times (d+1)}$. By the Caratheodory's Theorem, there must exist at most $(d+1)^2 + 1$ matrices $(a_i, \mathbf{b}_i) \cdot (a_i, \mathbf{b}_i)^\top$ whose convex hull also contains $\frac{1}{n} \cdot (A')^\top A'$. Then $\frac{1}{n} \cdot (A')^\top A'$ can be represented as a linear combination of these matrices, and hence, the sub-matrix $C \in \mathbb{R}^{((d+1)^2+1) \times (d+1)}$ exists.

Algorithm 1 of [33] shows how to directly construct such a matrix $C$. However, the running time is $O(n^2d^2)$ which is undesirable. To accelerate the running time, Jubran et al. [33] apply the following idea.

1. For each $i \in [n]$, set $p_i \in \mathbb{R}^{(d+1)^2}$ as the concatenation of the $(d+1)^2$ entries of $(a_i, \mathbf{b}_i) \cdot (a_i, \mathbf{b}_i)^\top$. Let $P$ be the collection of these points $p_i$. Then our objective is reduced to finding a subset $S \subseteq P$ of size $(d+1)^2 + 1$ such that the convex hull of $S$ contains $\overline{P} = \frac{1}{n} \cdot \sum_{i \in [n]} p_i$.

2. Compute a balanced partition $P_1, \ldots, P_k$ of $P$ into $k = 3(d+1)^2$ clusters of roughly the same size. By the Caratheodory's Theorem, there must exist at most $(d+1)^2 + 1$ partitions $P_i$ such that the convex hull of their union contains $\overline{P}$. The main issue is how to these partitions $P_i$ efficiently.

3. To address this issue, Jubran et al. [33] compute a sketch for each partition $P_i$ including its size $|P_i|$ and the weighted mean

$$u_i := \frac{1}{|P_i|} \cdot \sum_{j \in P_i} p_j.$$

The construction of sketches costs $O(nd^2)$ time. The key observation is that there exists a set $S$ of at most $(d+1)^2 + 1$ points $u_i$ such that the convex hull of their union contains $\overline{P}$ by the Caratheodory's Theorem. Moreover, the corresponding partitions $P_i$ of these $u_i$ are what we need – the convex hull of $\bigcup_{i \in [n]: u_i \in S} P_i$ contains $\overline{P}$. Note that the construction of

$S$ costs $O\left(k^2\left((d+1)^2\right)^2\right) = O(d^8)$ time. Overall, it costs $O(nd^2 + d^8)$ time to obtain the collection $\bigcup_{i\in[n]:u_i\in S} P_i$ whose convex hull contains $\overline{P}$.

4. We repeat the above procedure over $\bigcup_{i\in[n]:u_i\in S} P_i$ until obtaining an accurate coreset of size $(d+1)^2 + 1$. By the value of $k$, we note that

$$\left| \bigcup_{i\in[n]:u_i\in S} P_i \right| \le n/2,$$

i.e., we half the size of the input set by an iteration. Thus, there are at most $\log(n/d)$ iterations and the overall running time is

$$\sum_{i=0}^{\log n} \frac{O(nd^2)}{2^i} + O(d^8)\cdot\log(n/d) = O\left(nd^2 + d^8\log(n/d)\right).$$

# D   Missing proofs in Section 4

In this section, we complete the proofs for GLSE. Recall that the parameter space $\mathcal{P}_\lambda = \mathbb{R}^d \times B^q_{1-\lambda}$ for some constant $\lambda \in (0,1)$ where

$$B^q_{1-\lambda} = \left\{\rho \in \mathbb{R}^q : \|\rho\|_2^2 \le 1-\lambda\right\}.$$

Also recall that

$$Z^{(G,q)} = \left\{z_{it} = (x_{i,\max\{1,t-1\}}, y_{i,\max\{1,t-1\}}, \ldots x_{it}, y_{it}) : (i,t) \in [N] \times [T]\right\}.$$

Given two integers $a, b \ge 1$, denote $T(a,b)$ to be the computation time of a column basis of a matrix in $\mathbb{R}^{a\times b}$. For instance, a column basis of a matrix in $\mathbb{R}^{a\times b}$ can be obtained by computing its SVD decomposition, which costs $O(\min\left\{a^2 b, ab^2\right\})$ time by [14].

## D.1   Proof of Lemma 4.3: Upper bounding the pseudo-dimension

Our proof idea is similar to that in [37]. For preparation, we need the following lemma which is proposed to bound the pseudo-dimension of feed-forward neural networks.

**Lemma D.1 (Restatement of Theorem 8.14 of [2])** *Let $(\mathcal{X}, u, \mathcal{P}, f)$ be a given query space where $f_x(\zeta) \in \{0,1\}$ for any $x \in \mathcal{X}$ and $\zeta \in \mathcal{P}$, and $\mathcal{P} \subseteq \mathbb{R}^m$. Suppose that $f$ can be computed by an algorithm that takes as input the pair $(x,\zeta) \in \mathcal{X} \times \mathcal{P}$ and returns $f_x(\zeta)$ after no more than $l$ of the following operations:*

- *the arithmetic operations $+, -, \times$, and $/$ on real numbers.*
- *jumps conditioned on $>, \ge, <, \le, =$, and $\ne$ comparisons of real numbers, and*
- *output 0,1.*

*Then the pseudo-dimension of $(\mathcal{X}, u, \mathcal{P}, f)$ is at most $O(ml)$.*

Note that the above lemma requires that the range of functions $f_x$ is $[0,1]$. We have the following lemma which can help extend this range to $\mathbb{R}$.

**Lemma D.2 (Restatement of Lemma 4.1 of [53])** *Let $(\mathcal{X}, u, \mathcal{P}, f)$ be a given query space. Let $g_x : \mathcal{P} \times \mathbb{R} \to \{0,1\}$ be the indicator function satisfying that for any $x \in \mathcal{X}$, $\zeta \in \mathcal{P}$ and $r \in \mathbb{R}$,*

$$g_x(\zeta, r) = I\left[u(x)\cdot f(x,\zeta) \ge r\right].$$

*Then the pseudo-dimension of $(\mathcal{X}, u, \mathcal{P}, f)$ is precisely the pseudo-dimension of the query space $(\mathcal{X}, u, \mathcal{P} \times \mathbb{R}, g_f)$.*

Now we are ready to prove Lemma 4.3.

**Proof:** [Proof of Lemma 4.3] Fix a weight function $u : [N] \times [T] \to \mathbb{R}_{\geq 0}$. For every $(i,t) \in [N] \times [T]$, let $g_{it} : \mathcal{P}_\lambda \times \mathbb{R}_{\geq 0} \to \{0,1\}$ be the indicator function satisfying that for any $\zeta \in \mathcal{P}_\lambda$ and $r \in \mathbb{R}_{\geq 0}$,

$$g_{it}(\zeta, r) := I \left[ u(i,t) \cdot \psi_{it}^{(G,q)}(\zeta) \geq r \right].$$

We consider the query space $(Z^{(G,q)}, u, \mathcal{P}_\lambda \times \mathbb{R}_{\geq 0}, g)$. By the definition of $\mathcal{P}_\lambda$, the dimension of $\mathcal{P}_\lambda \times \mathbb{R}_{\geq 0}$ is $m = q + 1 + d$. By the definition of $\psi_{it}^{(G,q)}$, $g_{it}$ can be calculated using $l = O(qd)$ operations, including $O(qd)$ arithmetic operations and a jump. Pluging the values of $m$ and $l$ in Lemma D.1, the pseudo-dimension of $(Z^{(G,q)}, u, \mathcal{P}_\lambda \times \mathbb{R}_{\geq 0}, g)$ is $O((q+d)qd)$. Then by Lemma D.2, we complete the proof. $\square$

### D.2 Proof of Lemma 4.4: Bounding the total sensitivity

We prove Lemma 4.4 by relating sensitivities between GLSE and OLSE. For preparation, we give the following lemma that upper bounds the total sensitivity of OLSE. Recall that we denote $T(a,b)$ to be the computation time of a column basis of a matrix in $\mathbb{R}^{a \times b}$.

**Lemma D.3 (Total sensitivity of OLSE)** *Function $s^{(O)} : [N] \times [T] \to \mathbb{R}_{\geq 0}$ of Algorithm 1 satisfies that for any $(i,t) \in [N] \times [T]$,*

$$s^{(O)}(i,t) \geq \sup_{\beta \in \mathbb{R}^d} \frac{\psi_{it}^{(O)}(\beta)}{\psi^{(O)}(\beta)}, \tag{4}$$

*and $\mathcal{G}^{(O)} := \sum_{(i,t) \in [N] \times [T]} s^{(O)}(i,t)$ satisfying $\mathcal{G}^{(O)} \leq d + 1$. Moreover, the construction time of function $s^{(O)}$ is $T(NT, d+1) + O(NTd)$.*

**Proof:** The proof idea comes from [52]. By Line 3 of Algorithm 1, $A \subseteq \mathbb{R}^{NT \times d'}$ is a matrix whose columns form a unit basis of the column space of $Z$. We have $d' \leq d + 1$ and hence $\|A\|_2^2 = d' \leq d + 1$. Moreover, for any $(i,t) \in [N] \times [T]$ and $\beta' \in \mathbb{R}^{d'}$, we have

$$\|\beta'\|_2^2 \leq \|A\beta'\|_2^2,$$

Then by Cauchy-Schwarz and orthonormality of $A$, we have that for any $(i,t) \in [N] \times [T]$ and $\beta' \in \mathbb{R}^{d+1}$,

$$|z_{it}^\top \beta'|^2 \leq \|A_{iT-T+t}\|_2^2 \cdot \|Z\beta'\|_2^2, \tag{5}$$

where $A_{iT-T+t}$ is the $(iT - T + t)$-th row of $A$.

For each $(i,t) \in [N] \times [T]$, we let $s^{(O)}(i,t) := \|A_{iT-T+t}\|_2^2$. Then $\mathcal{G}^{(O)} = \|A\|_2^2 = d' \leq d + 1$. Note that constructing $A$ costs $T(NT, d+1)$ time and computing all $\|A_{iT-T+t}\|_2^2$ costs $O(NTd)$ time.

Thus, it remains to verify that $s^{(O)}(i,t)$ satisfies Inequality (4). For any $(i,t) \in [N] \times [T]$ and $\beta \in \mathbb{R}^d$, letting $\beta' = (\beta, -1)$, we have

$$
\begin{array}{lll}
\psi_{it}^{(O)}(\beta) = & |z_{it}^\top \beta'|^2 & \text{(Defn. of } \psi_{it}^{(O)}) \\
\leq & \|A_{iT-T+t}\|_2^2 \cdot \|Z\beta'\|_2^2 & \text{(Ineq. (5))} \\
= & \|A_{iT-T+t}\|_2^2 \cdot \psi^{(O)}(\beta). & \text{(Defn. of } \psi^{(O)})
\end{array}
$$

This completes the proof. $\square$

Now we are ready to prove Lemma 4.4.

**Proof:** [Proof of Lemma 4.4] For any $(i,t) \in [N] \times [T]$, recall that $s(i,t)$ is defined by

$$s(i,t) := \min$$
$$\left\{ 1, 2\lambda^{-1} \cdot \left( s^{(O)}(i,t) + \sum_{j=1}^{\min\{t-1,q\}} s^{(O)}(i,t-j) \right) \right\}.$$

We have that

$$\sum_{(i,t)\in[N]\times[T]} s(i,t)$$

$$\leq \sum_{(i,t)\in[N]\times[q]} 2\lambda^{-1}$$

$$\times \left(s^{(O)}(i,t) + \sum_{j=1}^{\min\{t-1,q\}} s^{(O)}(i,t-j)\right)$$

(by definition)

$$\leq 2\lambda^{-1}\cdot\sum_{(i,t)\in[N]\times[T]}(1+q)\cdot s^{(O)}(i,t)$$

$$\leq 2\lambda^{-1}(q+1)(d+1). \qquad \text{(Lemma D.3)}$$

Hence, the total sensitivity $\mathcal{G} = O(\lambda^{-1}qd)$. By Lemma D.3, it costs $T(NT,d+1) + O(NTd)$ time to construct $s^{(O)}$. We also know that it costs $O(NTq)$ time to compute function $s$. Since $T(NT,d+1) = O(NTd^2)$, this completes the proof for the running time.

Thus, it remains to verify that $s(i,t)$ satisfies that

$$s(i,t) \geq \sup_{\zeta\in\mathcal{P}} \frac{\psi_{it}^{(G,q)}(\zeta)}{\psi^{(G,q)}(\zeta)}.$$

Since $\sup_{\beta\in\mathbb{R}^d} \frac{\psi_{it}^{(O)}(\beta)}{\psi^{(O)}(\beta)} \leq 1$ always holds, we only need to consider the case that

$$s(i,t) = 2\lambda^{-1}\cdot\left(s^{(O)}(i,t) + \sum_{j=1}^{\min\{t-1,q\}} s^{(O)}(i,t-j)\right).$$

We first show that for any $\zeta = (\beta,\rho)\in\mathcal{P}_\lambda$,

$$\psi^{(G,q)}(\zeta) \geq \lambda\cdot\psi^{(O)}(\beta). \tag{6}$$

Given an autocorrelation vector $\rho\in\mathbb{R}^q$, the induced covariance matrix $\Omega_\rho$ satisfies that $\Omega_\rho^{-1} = P_\rho^\top P_\rho$ where

$$P_\rho = \begin{bmatrix} \sqrt{1-\|\rho\|_2^2} & 0 & 0 & \ldots & \ldots & \ldots & 0 \\ -\rho_1 & 1 & 0 & \ldots & \ldots & \ldots & 0 \\ -\rho_2 & -\rho_1 & 1 & \ldots & \ldots & \ldots & 0 \\ \ldots & \ldots & \ldots & \ldots & \ldots & \ldots & \ldots \\ 0 & 0 & 0 & -\rho_q & \ldots & -\rho_1 & 1 \end{bmatrix}.$$

Then by Equation (7), the smallest eigenvalue of $P_\rho$ satisfies that

$$\begin{aligned} \lambda_{\min} &= \sqrt{1-\|\rho\|_2^2} & \text{(Defn. of } P_\rho\text{)} \\ &\geq \sqrt{\lambda}. & (\rho\in B_{1-\lambda}^q) \end{aligned} \tag{7}$$

Also we have

$$\begin{aligned} \psi^{(G,q)}(\zeta) &= \sum_{i\in[N]} (y_i - X_i\beta)^\top \Omega_\rho^{-1}(y_i - X_i\beta) \\ &\qquad \text{(Program (GLSE))} \\ &= \sum_{i\in[N]} \|P_\rho(y_i - X_i\beta)\|_2^2 \\ &\qquad (P_\rho^\top P_\rho = \Omega_\rho^{-1}) \\ &\geq \sum_{i\in[N]} \lambda\cdot\|(y_i - X_i\beta)\|_2^2 \\ &\qquad \text{(Ineq. (7))} \\ &= \lambda\cdot\psi^{(O)}(\beta), \\ &\qquad \text{(Defns. of } \psi^{(O)}\text{)} \end{aligned}$$

which proves Inequality (6). We also claim that for any $(i,t) \in [N] \times [T]$,

$$\psi_{it}^{(G,q)}(\zeta) \leq 2 \cdot \left( \psi_{it}^{(O)}(\beta) + \sum_{j=1}^{\min\{t-1,q\}} \psi_{i,t-j}^{(O)}(\beta) \right). \tag{8}$$

This trivially holds for $t = 1$. For $t \geq 2$, this is because

$$\psi_{it}^{(G,q)}(\zeta)$$
$$= \left( (y_{it} - x_{it}^\top \beta) - \sum_{j=1}^{\min\{t-1,q\}} \rho_j \cdot (y_{i,t-j} - x_{i,t-j}^\top \beta) \right)^2$$
$$(t \geq 2)$$
$$\leq \left( 1 + \sum_{j=1}^{\min\{t-1,q\}} \rho_j^2 \right)$$
$$\times \left( (y_{it} - x_{it}^\top \beta)^2 + \sum_{j=1}^{\min\{t-1,q\}} (y_{i,t-j} - x_{i,t-j}^\top \beta)^2 \right)$$
$$\text{(Cauchy-Schwarz)}$$
$$= 2 \cdot \left( \psi_{it}^{(O)}(\beta) + \sum_{j=1}^{\min\{t-1,q\}} \psi_{i,t-j}^{(O)}(\beta) \right).$$
$$(\|\rho\|_2^2 \leq 1)$$

Now combining Inequalities (6) and (8), we have that for any $\zeta = (\beta, \rho) \in \mathcal{P}_\lambda$,

$$\frac{\psi_{it}^{(G,q)}(\zeta)}{\psi^{(G,q)}(\zeta)} \leq \frac{2 \cdot \left( \psi_{it}^{(O)}(\beta) + \sum_{j=1}^{\min\{t-1,q\}} \psi_{i,t-j}^{(O)}(\beta) \right)}{\lambda \cdot \psi^{(O)}(\beta)}$$
$$\leq 2\lambda^{-1} \cdot \left( s^{(O)}(i,t) + \sum_{j=1}^{\min\{t-1,q\}} s^{(O)}(i,t-j) \right)$$
$$= s(i,t).$$

This completes the proof. □

# E   Missing proofs in Section 5

In this section, we complete the proofs for $\text{GLSE}_k$.

## E.1   Proof overview

We first give a proof overview for summarization.

**Proof overview of Theorem 5.2.**   For $\text{GLSE}_k$, we propose a two-staged framework (Algorithm 2): first sample a collection of individuals and then run CGLSE on every selected individuals. By Theorem 4.1, each subset $J_{S,i}$ at the second stage is of size $\text{poly}(q,d)$. Hence, we only need to upper bound the size of $I_S$ at the first stage. By a similar argument as that for GLSE, we can define the pseudo-dimension of $\text{GLSE}_k$ and upper bound it by $\text{poly}(k,q,d)$, and hence, the main difficulty is to upper bound the total sensitivity of $\text{GLSE}_k$. We show that the gap between the individual regression objectives of $\text{GLSE}_k$ and $\text{OLSE}_k$ ($\text{GLSE}_k$ with $q = 0$) with respect to the same $(\beta^{(1)}, \ldots, \beta^{(k)})$ is at most $\frac{2(q+1)}{\lambda}$, which relies on $\psi_i^{(G,q)}(\zeta) \geq \lambda \cdot \psi_i^{(O)}(\beta)$ and an observation that for any $\zeta = (\beta^{(1)}, \ldots, \beta^{(k)}, \rho^{(1)}, \ldots, \rho^{(k)}) \in \mathcal{P}^k$, $\psi_i^{(G,q,k)}(\zeta) \leq 2(q+1) \cdot \min_{l \in [k]} \psi_i^{(O)}(\beta^{(l)})$. Thus, it suffices to provide an upper bound of the total sensitivity for $\text{OLSE}_k$. We claim that the maximum influence of individual $i$ is at most $\frac{u_i}{u_i + \sum_{j \neq i} \ell_j}$ where $u_i$ is the largest eigenvalue of $(Z^{(i)})^\top Z^{(i)}$ and $\ell_j$ is the smallest eigenvalue of $(Z^{(j)})^\top Z^{(j)}$. This fact comes from the following observation: $\min_{l \in [k]} \|Z^{(i)}(\beta^{(l)}, -1)\|_2^2 \leq \frac{u_i}{\ell_j} \cdot \min_{l \in [k]} \|Z^{(j)}(\beta^{(l)}, -1)\|_2^2$, and results in an upper bound $M$ of the total sensitivity for $\text{OLSE}_k$ since $\sum_{i \in [N]} \frac{u_i}{u_i + \sum_{j \neq i} \ell_j} \leq \frac{\sum_{i \in [N]} u_i}{\sum_{j \in [N]} \ell_j} \leq M$.

**Proof overview of Theorem 5.4.**   For $\text{GLSE}_k$, we provide a lower bound $\Omega(N)$ of the coreset size by constructing an instance in which any 0.5-coreset should contain observations from all individuals. Note that we consider $T = 1$ which reduces to an instance with cross-sectional data. Our instance

is to let $x_{i1} = (4^i, \frac{1}{4^i})$ and $y_{i1} = 0$ for all $i \in [N]$. Then letting $\zeta^{(i)} = (\beta^{(1)}, \beta^{(2)}, \rho^{(1)}, \rho^{(2)})$ where $\beta^{(1)} = (\frac{1}{4^i}, 0)$, $\beta^{(2)} = (0, 4^i)$ and $\rho^{(1)} = \rho^{(2)} = 0$, we observe that $\psi^{(G,q,k)}(\zeta^{(i)}) \approx \psi_i^{(G,q,k)}(\zeta^{(i)})$. Hence, all individuals should be contained in the coreset such that regression objectives with respect to all $\zeta^{(i)}$ are approximately preserved.

## E.2 Proof of Theorem 5.2: Upper bound for GLSE$_k$

The proof of Theorem 5.2 relies on the following two theorems. The first theorem shows that $I_S$ of Algorithm 2 is an $\frac{\varepsilon}{3}$-coreset of $(Z^{G,q,k}, \mathcal{P}_\lambda^k, \psi^{(G,q,k)})$. The second one is a reduction theorem that for each individual in $I_S$ constructs an $\varepsilon$-coreset $J_{S,i}$.

**Theorem E.1 (Coresets of $(Z^{G,q,k}, \mathcal{P}_\lambda^k, \psi^{(G,q,k)})$)** *For any given $M$-bounded observation matrix $X \in \mathbb{R}^{N \times T \times d}$ and outcome matrix $Y \in \mathbb{R}^{N \times T}$, constant $\varepsilon, \delta, \lambda \in (0,1)$ and integers $q, k \geq 1$, with probability at least 0.95, the weighted subset $I_S$ of Algorithm 2 is an $\frac{\varepsilon}{3}$-coreset of the query space $(Z^{G,q,k}, \mathcal{P}_\lambda^k, \psi^{(G,q,k)})$, i.e., for any $\zeta = (\beta^{(1)}, \ldots, \beta^{(k)}, \rho^{(1)}, \ldots, \rho^{(k)}) \in \mathcal{P}_\lambda^k$,*

$$\sum_{i \in I_S} w'(i) \cdot \psi_i^{(G,q,k)}(\zeta) \in (1 \pm \frac{\varepsilon}{3}) \cdot \psi^{(G,q,k)}(\zeta). \tag{9}$$

*Moreover, the construction time of $I_S$ is*

$$N \cdot \mathsf{SVD}(T, d+1) + O(N).$$

We defer the proof of Theorem E.1 later.

**Theorem E.2 (Reduction from coresets of $(Z^{G,q,k}, \mathcal{P}_\lambda^k, \psi^{(G,q,k)})$ to coresets for GLSE$_k$)**
*Suppose that the weighted subset $I_S$ of Algorithm 2 is an $\frac{\varepsilon}{3}$-coreset of the query space $(Z^{G,q,k}, \mathcal{P}_\lambda^k, \psi^{(G,q,k)})$. Then with probability at least 0.95, the output $(S, w)$ of Algorithm 2 is an $\varepsilon$-coreset for GLSE$_k$.*

**Proof:** [Proof of Theorem E.2] Note that $S$ is an $\varepsilon$-coreset for GLSE$_k$ if Inequality (9) holds and for all $i \in [N]$, $J_{S,i}$ is an $\frac{\varepsilon}{3}$-coreset of $((Z^{(i)})^{(G,q)}, \mathcal{P}_\lambda, \psi^{(G,q)})$. By condition, we assume Inequality (9) holds. By Line 6 of Algorithm 2, the probability that every $J_{S,i}$ is an $\frac{\varepsilon}{3}$-coreset of $((Z^{(i)})^{(G,q)}, \mathcal{P}_\lambda, \psi^{(G,q)})$ is at least

$$1 - \Gamma \cdot \frac{1}{20\Gamma} = 0.95,$$

which completes the proof. $\qquad\square$

Observe that Theorem 5.2 is a direct corollary of Theorems E.1 and E.2.

**Proof:** Combining Theorems E.1 and E.2, $S$ is an $\varepsilon$-coreset of $(Z^{G,q,k}, \mathcal{P}_\lambda^k, \psi^{(G,q,k)})$ with probability at least 0.9. By Theorem 4.1, the size of $S$ is

$$\Gamma \cdot O\left(\varepsilon^{-2}\lambda^{-1}qd\left(\max\{q^2d, qd^2\} \cdot \log\frac{d}{\lambda} + \log\frac{\Gamma}{\delta}\right)\right),$$

which satisfies Theorem 5.2 by pluging in the value of $\Gamma$.

For the running time, it costs $N \cdot \mathsf{SVD}(T, d+1)$ to compute $I_S$ by Theorem E.1. Moreover, by Line 3 of Algorithm 2, we already have the SVD decomposition of $Z^{(i)}$ for all $i \in [N]$. Then it only costs $O(T(q+d))$ to apply CGLSE for each $i \in I_S$ in Line 8 of Algorithm 2. Then it costs $O(NT(q+d))$ to construct $S$. This completes the proof of the running time. $\qquad\square$

**Proof of Theorem E.1: $I_S$ is a coreset of $(Z^{(G,q,k)}, \mathcal{P}_\lambda^k, \psi^{(G,q,k)})$.** It remains to prove Theorem E.1. Note that the construction of $I_S$ applies the Feldman-Langberg framework. The analysis is similar to Section D in which we provide upper bounds for both the total sensitivity and the pseudo-dimension.

We first discuss how to bound the total sensitivity of $(Z^{(G,q,k)}, \mathcal{P}^k, \psi^{(G,q,k)})$. Similar to Section D.2, the idea is to first bound the total sensitivity of $(Z^{(G,0,k)}, \mathcal{P}^k, \psi^{(G,0,k)})$ – we call it the query space of OLSE$_k$ whose covariance matrices of all individuals are identity matrices.

**Lemma E.3 (Total sensitivity of OLSE$_k$)** *Function $s^{(O)} : [N] \to \mathbb{R}_{\geq 0}$ of Algorithm 2 satisfies that for any $i \in [N]$,*

$$s^{(O)}(i) \geq \sup_{\beta^{(1)},\ldots,\beta^{(k)}\in\mathbb{R}^d} \frac{\min_{l\in[k]} \psi_i^{(O)}(\beta^{(l)})}{\sum_{i'\in[N]} \min_{l\in[k]} \psi_{i'}^{(O)}(\beta^{(l)})}, \quad (10)$$

*and $\mathcal{G}^{(O)} := \sum_{i\in[N]} s^{(O)}(i)$ satisfying that $\mathcal{G}^{(O)} = O(M)$. Moreover, the construction time of function $s^{(O)}$ is*

$$N \cdot \mathsf{SVD}(T, d+1) + O(N).$$

**Proof:** For every $i \in [N]$, recall that $Z^{(i)} \in \mathbb{R}^{T\times(d+1)}$ is the matrix whose $t$-th row is $z_t^{(i)} = (x_{it}, y_{it}) \in \mathbb{R}^{d+1}$ for all $t \in [T]$. By definition, we have that for any $\beta \in \mathbb{R}^d$,

$$\psi_i^{(O)}(\beta) = \|Z^{(i)}(\beta, -1)\|_2^2.$$

Thus, by the same argument as in Lemma D.3, it suffices to prove that for any matrix sequences $Z^{(1)}, \ldots, Z^{(N)} \in \mathbb{R}^{T\times(d+1)}$,

$$s^{(O)}(i) \geq \sup_{\beta^{(1)},\ldots,\beta^{(k)}\in\mathbb{R}^d}$$
$$\frac{\min_{l\in[k]} \|Z^{(i)}(\beta^{(l)}, -1)\|_2^2}{\sum_{i'\in[N]} \min_{l\in[k]} \|Z^{(i')}(\beta^{(l)}, -1)\|_2^2}. \quad (11)$$

For any $\beta^{(1)}, \ldots, \beta^{(k)} \in \mathbb{R}^d$ and any $i \neq j \in [N]$, letting $l^\star = \arg\min_{l\in[k]} \|Z^{(j)}(\beta^{(l)}, -1)\|_2^2$, we have

$$\min_{l\in[k]} \|Z^{(i)}(\beta^{(l)}, -1)\|_2^2$$
$$\leq \quad \|Z^{(i)}(\beta^{(l^\star)}, -1)\|_2^2$$
$$\leq \quad u_i \cdot (\|\beta^{(l^\star)}\|_2^2 + 1) \qquad\qquad \text{(Defn. of } u_i)$$
$$\leq \quad \frac{u_i}{\ell_j} \cdot \|Z^{(j)}(\beta^{(l^\star)}, -1)\|_2^2 \qquad\qquad \text{(Defn. of } \ell_i)$$
$$= \quad \frac{u_i}{\ell_j} \cdot \min_{l\in[k]} \|Z^{(j)}(\beta^{(l)}, -1)\|_2^2. \qquad\qquad \text{(Defn. of } l^\star)$$

Thus, we directly conclude that

$$\frac{\min_{l\in[k]} \|Z^{(i)}(\beta^{(l)}, -1)\|_2^2}{\sum_{i'\in[N]} \min_{l\in[k]} \|Z^{(i')}(\beta^{(l)}, -1)\|_2^2}$$
$$\leq \quad \frac{\min_{l\in[k]} \|Z^{(i)}(\beta^{(l)}, -1)\|_2^2}{\left(1 + \sum_{i'\neq i} \frac{\ell_{i'}}{u_i}\right) \cdot \min_{l\in[k]} \|Z^{(i)}(\beta^{(l)}, -1)\|_2^2}$$
$$= \quad \frac{u_i}{u_i + \sum_{i'\neq i} \ell_{i'}}$$
$$= \quad s^{(O)}(i).$$

Hence, Inequality (11) holds. Moreover, since the input dataset is $M$-bounded, we have

$$\mathcal{G}^{(O)} \leq \sum_{i\in[N]} \frac{u_i}{\sum_{i'\in[N]} \ell_{i'}} \leq M,$$

which completes the proof of correctness.

For the running time, it costs $N \cdot \mathsf{SVD}(T, d+1)$ to compute SVD decompositions for all $Z^{(i)}$. Then it costs $O(N)$ time to compute all $u_i$ and $\ell_i$, and hence costs $O(N)$ time to compute sensitivity functions $s^{(O)}$. Thus, we complete the proof. $\qquad\square$

Note that by the above argument, we can also assume

$$\sum_{i \in [N]} \frac{u_i}{u_i + \sum_{i' \neq i} \ell_{i'}} \leq M,$$

which leads to the same upper bound for the total sensitivity $\mathcal{G}^{(O)}$. Now we are ready to upper bound the total sensitivity of $(Z^{(G,q,k)}, \mathcal{P}^k, \psi^{(G,q,k)})$.

**Lemma E.4 (Total sensitivity of GLSE$_k$)** *Function* $s : [N] \to \mathbb{R}_{\geq 0}$ *of Algorithm 2 satisfies that for any* $i \in [N]$,

$$s(i) \geq \sup_{\zeta \in \mathcal{P}_\lambda^k} \frac{\psi_i^{(G,q,k)}(\zeta)}{\psi^{(G,q,k)}(\zeta)}, \tag{12}$$

*and* $\mathcal{G} := \sum_{i \in [N]} s(i)$ *satisfying that* $\mathcal{G} = O(\frac{qM}{\lambda})$. *Moreover, the construction time of function* $s$ *is*

$$N \cdot \mathsf{SVD}(T, d+1) + O(N).$$

**Proof:** Since it only costs $O(N)$ time to construct function $s$ if we have $s^{(O)}$, we prove the construction time by Lemma E.3.

Fix $i \in [N]$. If $s(i) = 1$ in Line 4 of Algorithm 2, then Inequality (12) trivially holds. Then we assume that $s(i) = \frac{2(q+1)}{\lambda} \cdot s^{(O)}(i)$. We first have that for any $i \in [N]$ and any $\zeta \in \mathcal{P}_\lambda^k$,

$$\psi_i^{(G,q,k)}(\zeta)$$

$$= \min_{l \in [k]} \sum_{t \in [T]} \psi_{it}^{(G,q)}(\beta^{(l)}, \rho^{(l)}) \qquad \text{(Defn. 2.4)}$$

$$\geq \min_{l \in [k]} \sum_{t \in [T]} \lambda \cdot \psi_{it}^{(O)}(\beta^{(l)}) \qquad \text{(Ineq. (6))}$$

$$= \lambda \cdot \min_{l \in [k]} \psi_i^{(O)}(\beta^{(l)}). \qquad \text{(Defn. of } \psi_i^{(O)})$$

which directly implies that

$$\psi^{(G,q,k)}(\zeta) \geq \lambda \cdot \sum_{i' \in [N]} \min_{l \in [k]} \psi_{i'}^{(O)}(\beta^{(l)}). \tag{13}$$

We also note that for any $(i, t) \in [N] \times [T]$ and any $(\beta, \rho) \in \mathcal{P}_\lambda$,

$$\psi_{it}^{(G,q)}(\beta, \rho)$$

$$\leq \left( (y_{it} - x_{it}^\top \beta) - \sum_{j=1}^{\min\{t-1,q\}} \rho_j \cdot (y_{i,t-j} - x_{i,t-j}^\top \beta) \right)^2 \qquad \text{(Defn. of } \psi_{it}^{(G,q)})$$

$$\leq \left( 1 + \sum_{j=1}^{\min\{t-1,q\}} \rho_j^2 \right) \times \left( (y_{it} - x_{it}^\top \beta)^2 + \sum_{j=1}^{\min\{t-1,q\}} (y_{i,t-j} - x_{i,t-j}^\top \beta)^2 \right) \quad \text{(Cauchy-Schwarz)}$$

$$\leq 2 \left( (y_{it} - x_{it}^\top \beta)^2 + \sum_{j=1}^{\min\{t-1,q\}} (y_{i,t-j} - x_{i,t-j}^\top \beta)^2 \right). \qquad (\|\rho\|_2^2 \leq 1)$$

Hence, we have that

$$\frac{1}{2} \cdot \psi_{it}^{(G,q)}(\beta, \rho) \leq (y_{it} - x_{it}^\top \beta)^2 + \sum_{j=1}^{\min\{t-1,q\}} (y_{i,t-j} - x_{i,t-j}^\top \beta)^2. \tag{14}$$

This implies that

$$\psi_i^{(G,q,k)}(\zeta)$$

$$= \min_{l \in [k]} \sum_{t \in [T]} \psi_{it}^{(G,q)}(\beta^{(l)}, \rho^{(l)}) \qquad \text{(Defn. 2.4)}$$

$$\leq \min_{l \in [k]} \sum_{t \in [T]} 2 \times \left( (y_{it} - x_{it}^\top \beta)^2 + \sum_{j=1}^{\min\{t-1,q\}} (y_{i,t-j} - x_{i,t-j}^\top \beta)^2 \right) \qquad \text{(Ineq. (14))} \tag{15}$$

$$\leq 2(q+1) \cdot \min_{l \in [k]} \sum_{t \in [T]} \psi_{it}^{(O)}(\beta^{(l)})$$

$$= 2(q+1) \cdot \min_{l \in [k]} \psi_i^{(O)}(\beta^{(l)}). \qquad \text{(Defn. of } \psi_i^{(O)})$$

Thus, we have that for any $i \in [N]$ and $\zeta \in \mathcal{P}_\lambda^k$,

$$\frac{\psi_i^{(G,q,k)}(\zeta)}{\psi^{(G,q,k)}(\zeta)} \leq \frac{2(q+1) \cdot \min_{l \in [k]} \psi_i^{(O)}(\beta^{(l)})}{\lambda \cdot \sum_{i \in [N]} \min_{l \in [k]} \psi_i^{(O)}(\beta^{(l)})} \qquad \text{(Ineqs. (13) and (15))}$$

$$\leq \frac{2(q+1)}{\lambda} \cdot s^{(O)}(i) \qquad \text{(Lemma E.3)}$$

$$= s(i), \qquad \text{(by assumption)}$$

which proves Inequality (12). Moreover, we have that

$$\mathcal{G} = \sum_{i \in [N]} s(i) \leq \frac{2(q+1)}{\lambda} \cdot \mathcal{G}^{(O)} = O(\frac{qM}{\lambda}),$$

where the last inequality is from Lemma E.3. We complete the proof. $\qquad\square$

Next, we upper bound the pseudo-dimension of $\text{GLSE}_k$. The proof is similar to that of GLSE by applying Lemmas D.1 and D.2.

**Lemma E.5 (Pseudo-dimension of GLSE$_k$)** *The pseudo-dimension of any query space $(Z^{(G,q,k)}, u, \mathcal{P}_\lambda^k, \psi^{(G,q,k)})$ over weight functions $u : [N] \to \mathbb{R}_{\geq 0}$ is at most*
$$O\left(k^2 q^2 (q+d) d^2\right).$$

**Proof:** The proof idea is similar to that of Lemma 4.3. Fix a weight function $u : [N] \to \mathbb{R}_{\geq 0}$. For every $i \in [N]$, let $g_i : \mathcal{P}_\lambda^k \times \mathbb{R}_{\geq 0} \to \{0, 1\}$ be the indicator function satisfying that for any $\zeta = (\beta^{(1)}, \ldots, \beta^{(k)}, \rho^{(1)}, \ldots, \rho^{(k)}) \in \mathcal{P}_\lambda^k$ and $r \in \mathbb{R}_{\geq 0}$,

$$g_i(\zeta, r) := I\left[u(i) \cdot \psi_i^{(G,q,k)}(\zeta) \geq r\right]$$

$$= I\left[\forall l \in [k], \ u(i) \cdot \sum_{t \in [T]} \psi_{it}^{(G,q)}(\beta^{(l)}, \rho^{(l)}) \geq r\right].$$

We consider the query space $(Z^{(G,q,k)}, u, \mathcal{P}_\lambda^k \times \mathbb{R}_{\geq 0}, g)$. By the definition of $\mathcal{P}_\lambda^k$, the dimension of $\mathcal{P}_\lambda^k \times \mathbb{R}_{\geq 0}$ is $m = k(q+d)+1$. Also note that for any $(\beta, \rho) \in \mathcal{P}_\lambda$, $\psi_{it}^{(G,q)}(\beta, \rho)$ can be represented as a multivariant polynomial that consists of $O(q^2 d^2)$ terms $\rho_{c_1}^{b_1} \rho_{c_2}^{b_2} \beta_{c_3}^{b_3} \beta_{c_4}^{b_4}$ where $c_1, c_2 \in [q], c_3, c_4 \in [d]$ and $b_1, b_2, b_3, b_4 \in \{0, 1\}$. Thus, $g_i$ can be calculated using $l = O(kq^2 d^2)$ operations, including $O(kq^2 d^2)$ arithmetic operations and $k$ jumps. Pluging the values of $m$ and $l$ in Lemma D.1, the pseudo-dimension of $(Z^{(G,q,k)}, u, \mathcal{P}_\lambda^k \times \mathbb{R}_{\geq 0}, g)$ is $O\left(k^2 q^2 (q+d) d^2\right)$. Then by Lemma D.2, we complete the proof. $\qquad\square$

Combining with the above lemmas and Theorem **??**, we are ready to prove Theorem E.1.

**Proof:** [Proof of Theorem E.1] By Lemma E.4, the total sensitivity $\mathcal{G}$ of $(Z^{(G,q,k)}, \mathcal{P}_\lambda^k, \psi^{(G,q,k)})$ is $O(\frac{qM}{\lambda})$. By Lemma E.5, we can let $\dim = O\left(k^2 (q+d) q^2 d^2\right)$ which is an upper bound of the pseudo-dimension of every query space $(Z^{(G,q,k)}, u, \mathcal{P}_\lambda^k, \psi^{(G,q,k)})$ over weight functions $u : [N] \to \mathbb{R}_{\geq 0}$. Pluging the values of $\mathcal{G}$ and $\dim$ in Theorem **??**, we prove for the coreset size.

For the running time, it costs $N \cdot \text{SVD}(T, d+1) + O(N)$ time to compute the sensitivity function $s$ by Lemma E.4, and $O(N)$ time to construct $I_S$. This completes the proof. $\qquad\square$

### E.3 Proof of Theorem 5.4: Lower bound for GLSE$_k$

Actually, we prove a stronger version of Theorem 5.4 in the following. We show that both the coreset size and the total sensitivity of the query space $(Z^{(G,q,k)}, u, \mathcal{P}_\lambda^k, \psi^{(G,q,k)})$ may be $\Omega(N)$, even for the simple case that $T = 1$ and $d = k = 2$.

**Theorem E.6 (Size and sensitivity lower bound of GLSE$_k$)** *Let $T = 1$ and $d = k = 2$ and $\lambda \in (0, 1)$. There exists an instance $X \in \mathbb{R}^{N \times T \times d}$ and $Y \in \mathbb{R}^{N \times T}$ such that the total sensitivity*

$$\sum_{i \in [N]} \sup_{\zeta \in \mathcal{P}_\lambda^k} \frac{\psi_i^{(G,q,k)}(\zeta)}{\psi^{(G,q,k)}(\zeta)} = \Omega(N).$$

*and any 0.5-coreset of the query space $(Z^{(G,q,k)}, u, \mathcal{P}_\lambda^k, \psi^{(G,q,k)})$ should have size $\Omega(N)$.*

**Proof:** We construct the same instance as in [49]. Concretely, for $i \in [N]$, let $x_{i1} = (4^i, \frac{1}{4^i})$ and $y_{i1} = 0$. We claim that for any $i \in [N]$,

$$\sup_{\zeta \in \mathcal{P}_\lambda^k} \frac{\psi_i^{(G,q,k)}(\zeta)}{\psi^{(G,q,k)}(\zeta)} \geq \frac{1}{2}. \tag{16}$$

If the claim is true, then we complete the proof of the total sensitivity by summing up the above inequality over all $i \in [N]$. Fix $i \in [N]$ and consider the following $\zeta = (\beta^{(1)}, \beta^{(2)}, \rho^{(1)}, \rho^{(2)}) \in \mathcal{P}_\lambda^k$ where $\beta^{(1)} = (\frac{1}{4^i}, 0)$, $\beta^{(2)} = (0, 4^i)$ and $\rho^{(1)} = \rho^{(2)} = 0$. If $j \leq i$, we have

$$\psi_j^{(G,q,k)}(\zeta) = \min_{l \in [2]}(y_{i1} - x_{i1}^\top \beta^{(l)})^2$$

$$= \min\left\{\frac{1}{16^{j-i}}, \frac{1}{16^{i-j}}\right\}$$

$$= \frac{1}{16^{i-j}}.$$

Similarly, if $j > i$, we have

$$\psi_j^{(G,q,k)}(\zeta) = \min\left\{\frac{1}{16^{j-i}}, \frac{1}{16^{i-j}}\right\} = \frac{1}{16^{j-i}}.$$

By the above equations, we have

$$\psi^{(G,q,k)}(\zeta) = \sum_{j=1}^{i} \frac{1}{16^{i-j}} + \sum_{j=i+1}^{N} \frac{1}{16^{j-i}} < \frac{5}{4}. \tag{17}$$

Combining with the fact that $\psi_i^{(G,q,k)}(\zeta) = 1$, we prove Inequality (16).

For the coreset size, suppose $S \subseteq [N]$ together with a weight function $w : S \to \mathbb{R}_{\geq 0}$ is a 0.5-coreset of the query space $(Z^{(G,q,k)}, u, \mathcal{P}_\lambda^k, \psi^{(G,q,k)})$. We only need to prove that $S = [N]$. Suppose there exists some $i^\star \in S$ with $w(i^\star) > 2$. Letting $\zeta = (\beta^{(1)}, \beta^{(2)}, \rho^{(1)}, \rho^{(2)})$ where $\beta^{(1)} = (\frac{1}{4^{i^\star}}, 0)$, $\beta^{(2)} = (0, 4^{i^\star})$ and $\rho^{(1)} = \rho^{(2)} = 0$, we have that

$$\sum_{i \in S} w(i) \cdot \psi_i^{(G,q,k)}(\zeta) > \quad w(i^\star) \cdot \psi_{i^\star}^{(G,q,k)}(\zeta)$$

$$> \quad 2 \qquad\qquad (w(i^\star) > 2 \text{ and Defns. of } \zeta)$$

$$> \quad (1 + \frac{1}{2}) \cdot \frac{5}{4}$$

$$> \quad (1 + \frac{1}{2}) \cdot \psi^{(G,q,k)}(\zeta), \qquad\qquad \text{(Ineq. (17))}$$

which contradicts with the assumption of $S$. Thus, we have that for any $i \in S$, $w(i) \leq 2$. Next, by contradiction assume that $i^\star \notin S$. Again, letting $\zeta = (\beta^{(1)}, \beta^{(2)}, \rho^{(1)}, \rho^{(2)})$ where $\beta^{(1)} = (\frac{1}{4^{i^\star}}, 0)$, $\beta^{(2)} = (0, 4^{i^\star})$ and $\rho^{(1)} = \rho^{(2)} = 0$, we have that

$$\sum_{i \in S} w(i) \cdot \psi_i^{(G,q,k)}(\zeta) \leq \quad 2\left(\psi^{(G,q,k)}(\zeta) - \psi_{i^\star}^{(G,q,k)}(\zeta)\right)$$

$$(w(i) \leq 2)$$

$$\leq \quad 2(\frac{5}{4} - 1) \qquad\qquad \text{(Ineq. (17))}$$

$$\leq \quad (1 - \frac{1}{2}) \cdot 1$$

$$\leq \quad (1 - \frac{1}{2}) \cdot \psi^{(G,q,k)}(\zeta),$$

which contradicts with the assumption of $S$. This completes the proof.

$\square$

Figure 1: Boxplots of empirical errors for GLSE w.r.t. varying $\varepsilon$. **Uni** has higher average and maximum empirical errors than CGLSE.

# F   Other empirical results

We also provide boxplots for both synthetic and real-world datasets in Figure 1. The figures indicate that **Uni** has higher average and maximum empirical errors than CGLSE.

## Footnotes

[3] Here, we slightly abuse the notation by using $\psi_{it}^{(G,q)}(\zeta)$ instead of $\psi_{z_{it}}^{(G,q)}(\zeta)$.