[Reviews · NeurIPS 2020]

Review 1

Summary and Contributions: The paper deals with coresets (data summary obtained by subsampling and approximating the objective function for all queries) for least squares regression on panel data. In the usual "cross-sectional" setting the data consists of N individuals with d features each. Panel data extends this by introducing a time-series for each individual, consisting of the d features measured at T time steps. Moreover a correlation structure is introduced between the time steps to model dependencies over the time axes. The objective is to minimize the sum over all individuals of the least squares regressions subject to correlations between the time steps. The variables are thus the regression parameters as well as the defining parameters of the covariance structure. The problem is defined in this very general setting but the correlation structure studied is restricted to autocorrelations, resp. the autoregression model of order q, where each time step may depend on the q preceding time steps, and their q weights are the parameters subject to the optimization. The contributions are the first coreset constructions for this generalized regression setting obtained in the well-known in the sensitivity framework. The mere formulation of coresets is a bit more complicated than usual, since the terms in the sum of squares (sum f_i) do not correspond to the point x_i one by one. Instead each f_i may depend on up to q points x_j. The coreset is thus defined by subsampling and reweighting the functions f_i, and all points x_j that occur in the subsampled functions need to be stored. The paper spends a lot of space on making this formal. I think it could be explained much more concisely. The main complexity parameters to bound in teh sensitivity framework are the VC dimension of the functions to sample from and their "sensitivities" which bound their worst case importance for approximating the objective and their sum dominates the number of samples needed. The VC or pseudo dimension is bounded via a generic theorem from the literature by counting the number of parameters and the number of arithmetic operations to calculate the functions f_i. The sensitivities are bounded by a reduction to the sensitivities in the ordinary least squares (OLS) setting, which have well known results. The reduction is fairly straightforward in my opinion: The main challenge is to get rid of the correlations. For a lower bound the contribution of the covariance matrix is lower bounded by the smallest eigenvalue (lambda) times the contribution without covariance (OLS), note that lambda is assumed to be a constant in this paper by definition! The upper bound uses that the autocorrelation structure involves the square of a linear combination of the (non-squared) OLS terms. Now by Cauchy-Schwarz this is bounded by a linear combination of the squared OLS terms. This directly implies that the sensitivity of a function f_i is at most lambda times a linear combination of the corresponding OLS sensitivities, and the total sensitivity is simply the known small (independent of the input size) bound, times the maximal number of terms q. For other correlation structures this might be more challenging but the paper never considers others than autocorrelations. For the experimental data the standard generative model is used involving Gaussian errors from the linear model. It is known that in such a case the leverage scores, i.e. the OLS sensitivities are quite uniform (some paper of michael mahoney, I don't remember which). This might explain why the experimental results are not very convincing, see below. There is a further generalization of regression models to mixtures of linear regressions in the supplement. I didn't dig too much into that but it seems quite similar to existing work on Gaussian mixture models or k-means or alike. I think that the extension to panel regression is valuable and interesting as well as the k-clustered regression setting. On the other hand the results seem to be quite immediate from the literature, and the experiments are not very convincing (see weaknesses). Therefore I think that the current state is below the borderline. However I think that doing experiments on data that possesses less uniform errors would make a better selling point. Also discussing and bounding the sensitivities for more general or complicated covariances could improve the theoretical contributions. I also think that it's a pity to find those definitions for the sensitivity framework filling most of the paper while most of your own contributions appear only in the supplement, especially the k-clustered regression results. *** After Rebuttal: see additional feedback section below!!! ***

Strengths: - An important and natural extension of (ordinary) least squares regression - coreset sizes depend polynomially on 1/eps, d and q but have no dependence on N or T. - a further result on a mixture or k-clustering of least squares regressions (in analogy to coresets for mixtures of Gaussians or k-means)

Weaknesses: - spend too much on explaining the very general definitions of the sensitivity framework and fitting the problem into it. This could be done more concisely and defer the details to the appendix. - fairly straightforward reduction to the ordinary least squares regression case. - experimental results are not convincing, since worst case relative errors are within a factor of 2-3 to uniform subsampling even for small sample sizes, while average errors are often even worse (!!) than uniform subsampling (especially on the real-world data). It seems that the variance is reduced over uniform sampling but not the average behavior.

Correctness: yes all claims seem correct

Clarity: mostly yes, but it should move the sensitivity definitions into the appendix and move the own derivations and results into the main paper. Especially moving the results on k-clustering regression in the main body would improve the body of own contributions in the main paper.

Relation to Prior Work: yes, the generalized setting of panel regression is introduced and the challenges in constructing coresets are described and compared to well known constructions for the ordinary least squares case. further related work is given on coresets for ordinary and generalized regression.

Reproducibility: Yes

Additional Feedback: - in the beginning it is not clear that the parameters of the covariance is due to optimization (if it was fixed the reduction to OLS would be even simpler) - 76: \Omega(N) implies M is unavoidable? -> it should be Omega(M) right? - 79-80 the sentence is weird, missing words - last paragraph on page 2: here already the experimental results seem good compared to the full data but only slightly better than uniform sampling. -131: correlations between individuals -> should be between time steps, right? - 196: is -> in; compute -> computes; decides -> defines - Algo 1: "require ensure" -> "input output" - 252: missing "i" in index for x and y - 269: be more specific on the relevant hardware - have you experimented with larger values of q >1 ? - 287: average over what? - Tables 1 -> Table 1 - 299: add units (seconds) - 445-446: Equation (3) actually refers to the one above Eq (3) - it is weird that you "propose" theorems, then they are referenced from the literature, so others have proposed this before and then there are proofs that simply adapt the known theorems to the setting by direct application. - p. 15 what is T( x , y ) ? - (5) is simply by Cauchy-Schwarz and orthonormality of A, no need to look up the "well conditioned basis" definition from other literature Dear authors, thanks for your additional efforts and responses to our reviews. Your comment on GLSE_k made me read this part again in all details. Overall I decided to support your paper and raise my score significantly. However, I have some conditions: 1) incorporate carefully all points from your rebuttal and fix all issues that we raised, as you have promised. Regarding the new experiments add this reference: https://dl.acm.org/doi/10.5555/2789272.2831141 . It covers t-distributions with different parameters, including the special cases of Gaussian and Cauchy errors (note that the leverage scores are exactly your OLSE sensitivities!). 2) Don't waste so much space on tweaking the definitions of the framework. This can be done in a much more concise way and details can be moved to the appendix. Reference recent surveys on coresets, e.g. https://onlinelibrary.wiley.com/doi/full/10.1002/widm.1335 and https://link.springer.com/article/10.1007/s13218-017-0519-3. 3) Put GLSE_k back into the main paper! Emphasize the novelty of the "2-stage" coreset construction for the nested sums, and also the assumption of M-boundedness. Comment on the necessity of the latter and point out why M-boundedness for small M is natural to assume (e.g. what happens with M if the Matrices A(i) are drawn from normal distributions where the covariance has constant eigenvalues l_max...l_min?) Are there similarities to known results on coresets for mixtures of Gaussians regarding such nice conditioning assumptions? 4) (if time permits) can you add any experiments on GLSE_k? can you compare to mini-batchSGD? are there settings where coresets or SGD have advantages/deficiencies compared to one another? 5) fix some more minor issues in the appendix: - 713: the 2-stage coreset uses eps/2 on each stage. How do the errors propagate? I am missing calculations, e.g. (1+eps/2)^2 < (1+eps) does not hold... - 756: missing superscripts for the betas - 757: missing sum in denominator - the lower bound is akin to [39] which should be referenced in this context


Review 2

Summary and Contributions: The panel data problem is a generalization of linear regression and aims to minimize sum_i ||X(A_i x-b)||^2 over a matrix X and a vector b, where the sum is over large set of matrices A_1...A_n. Coresets for special cases are suggested: - For OLSE, coreset of size min{d/eps^2,d^2} - For GLSE, coreset of size d^2/eps^2 where d is a constant. - Generalization and Lower bounds for GLSE_k Justifications: - This is a very fundamental problem with many applications and generalizations in other fields. - There are many types of coresets in this paper, not just one - The theory is not trivial and it seems to contain many new ideas.

Strengths: The paper solves special versions of the problem min sum_i ||A_ix-b_i||^2 under different constraints. There are many results as stated above, including lower bounds, and the algorithms are interesting. The techniques are novel although not too surprising. I mainly liked the fact that there are few types of fundamental new coresets in the same paper. The paper has a potential to have high impact.

Weaknesses: - No open code as in related papers. - More experimental results are needed.

Correctness: I did not read all the proofs in the appendix, but the claims make sense.

Clarity: Yes, I liked to read it.

Relation to Prior Work: Yes.

Reproducibility: Yes

Additional Feedback:


Review 3

Summary and Contributions: The paper introduces coresets for the Generalized Least Square Estimators (GLSE) problem, and provides several algorithms for coresets for GLSE, supported by experiments. Summary: GLSE is a variant of multiobjective regression problem where observations are correlated (over time). Informally, coreset for a data set D with respect to a class of functions F is a small subset of D such that any function from F evaluates approximately equally on D and on S. OLSE is a variant of GLSE where the correlation matrix is identity. OLSE can be reduced to a single objective regression problem and (weak) coresets for OLSE problems have been studied in [27]. This paper makes direct generalizations of results from [27] and [9] to provide approximate and exact coresets for OLSE. Further the paper employs the well-known Feldman-Langberg (FL) framework to obtain a coreset for GLSE. It seems that FL is directly applicable and the main result in this paper is to bound the two parameters of FL framework: (1) total sensitivity and (2) pseudo-dimension of GLSE.

Strengths: Coresets for GLSE is a nice and novel result that extend results in [7] and [27] and, as the authors explain, can lead to future work in multi-objective regression such as logistic regression.

Weaknesses: The basic approaches in the paper seem to be fairly standard, to the best of my understanding. Both the algorithms and the bounds proofs seem quite similar to known techniques used in previous results. It might be the case that I missed some technical novelty. If there are novel ideas in bounding the pseudo-dimension or total sensitivity, the authors should carefully explain them in the introduction. The current explanation such as “Hence, to apply the FL framework, we need to upper bound the pseudo dimension and construct a sensitivity function.” and “The main idea is to apply the prior results [2, 43] “ strengthen my impression that the technical contribution is somewhat incremental.

Correctness: Due to the lack of time, I did not check the correctness of all technical claims but the overall approach makes sense to me.

Clarity: I think that the clarity can be improved by emphasizing the technical novelty in bounding the pseudo-dimension and total sensitivity. Also, in appendix B it is difficult to distinguish prior work from novel claims. Overall the paper is not easy to read.

Relation to Prior Work: see my comments above

Reproducibility: Yes

Additional Feedback:


Review 4

Summary and Contributions: In contrast to usual time series data, where we observe features for one individual (unit) across time, panel data (cross-sectional time series data) describes of a time series for several individuals (units) in parallel (i.e. same set of features recorded for each invidiual separately over time). In the regression problem on panel data we control variables that change over time but not across units: We seek for $min_{\beta, \Omega}\ (y_i-X_i\beta)^T\Omega^{-1}(y_i-X_i\beta)$ where $X_i$ is the time series for the i-th unit, and where the parameter $\Omega \in \R^{T \times T}$ is constraint to have eigenvalue at most 1. Two versions considered in the paper are: - In OLS (ordinary least squares), $\Omega = I_d$. - In GLS (generalised least sqares), $\Omega$ is defined by an autocorrelation function dependin on some parameter $\rho$. - Additionally, the paper mentions the GLS_k problem where entities are divided into k groups and (only) each group shares parameters -- but this is only described in the supplementary material. The idea of coresets/sketches is that, for a fixed problem (e.g. panelregression, clustering), to construct a smaller problem instance that behaves similar to the origial given problem instance. E.g. for clustering, coresets are usually subset of the given data points. Goal of the paper is to create a coreset that is independent in the number of entities (N) as well as time steps (T). For this, the paper shows how to apply a framework for coresets that was introduced by Langberg and Feldman, and previous results on near optimal coresets for least-squares regression by Boutsidis et al.. Hence the main step consists in deriving the so called sensitivity function. This yields an algorithm that constructs an eps-coreset of a size hat is independent on N and T, but depends on the success probability, eps, and an upper bound of the parameter $\rho$ (of the autocorrelation function), and runs in time linear in $N,T,q$ and quadratic in $d$. Additionally, the paper conducts some experiments with synthetic as well as real-world data.

Strengths: The paper presents a formal derivation (supplementary) of the sensitivity function for GLSE, which is not trivial. It shows how the abstract coreset framework of langberg and feldman can be applied to a new problem. It presents the first coreset construction for this problem with these guarantees (size independent of N and T). The paper tries to evaluate the performance of the methods on a real world data set.

Weaknesses: The way the GLSE problem is fitted into the framework by Langberg and Feldman is pretty straightforward. Yet, the paper e.g. introduces the generalisation of an query space, which does not add any real value in the main paper (~1 page). The necessary notation could as well have been introduced in Definitions 2.2 and 2.3 right away (which is what is basically done in ll.169 then - in addition to the original definition plus the coreset/query space definitions). The paper states that "When dealing with massive datasets, coresets have emerged as a valuable tool froma computational, storage and privacy perspective, as one needs to work with and share much smaller datasets." (ll. 3). But the only thing that can be done with these coresets is to solve the specific GLS problem (i.e. specific $\rho$, $q$,.. and all wrt. X) that the coreset was computed for. If one wants to e.g. test what happens with a different $q$, the coreset would need to be computed again. The experimental resusults are rather weak: With synthetic data, one can achieve lower error via sampling approx 4k points than with the proposed method using approx 800 points - and both approaches would take about the same time (6s vs 5s). For the real-world data, the uniform coreset is on avg. actually better (except for 819 sample size), which comparable standard deviations. The experimental section highlights the maximum empirical error in the comparison (boxplots would be more appropriate). Especially for the real-world data where on avg. the uniform method is outperforming the proposed method. The errors depend on how variables are scaled. A predefined significance level would help to interpret the results. (In the context of speeding up computations, one should keep in mind that the computational budget (in practice) might allow to compute uniform samples and to solve the problem on the uniform sample repeatedly (to achieve same results, at a small cost even without a clever algorithm)). (Also the experimental evaluation just aims to minimize the train error. This is the goal of the proposed method, but in practice, one would want to know how well the solution obtained via the coreset generates to unseen data. And then the difference between uniform and the proposed method might be smaller.)

Correctness: There are no full proofs in the main paper. Had a look at some of the proofs in the appendix, which look ok to me.

Clarity: The paper is clearly organized and clearly describes ideas and which ideas/algorithms where motivated by which papers, which is nice. It could be clearer on the limited applications regarding coresets, and on the interpretation of the experimental results which seem rather week (see weaknesses) and are presented in an irritating way.

Relation to Prior Work: Relation to prior work is clearly stated. The results are compared, and an overview e.g. on other coreset literature is included.

Reproducibility: Yes

Additional Feedback: Is $GLS_k$ used anywhere (useful in any real world context), or has it just been introduced e because we can? It is weird that GLS_k and existence of results are mentioned in the main part, but then no result is in the main paper. Maybe it would be better to mention this extension later and then also briefly describe what (not) changes in the main results (compared to the just presented results then). The maximum empirical error column is really irritating (especially since that column is marked in bold to underline the superiority of CGLSE). Instead of the table, boxplots incl. plots of outliers might be a better way to represent the results. It is not so easy to follow the notation sometimes: e.g. when there's a bigger gap between variable introduction and mentioning the variable name. E.g. "The $\rho$ variables make the objective function of GLSE in contrast to the cross-sectional data setting where objective functions only contain β and are convex." (ll. 79) - a $rho$ was mentioned last time in ll. 38, plus part of the sentence must be missing here? Based on author feedback and discussion: Thanks for your detailed feedback. I'll increase my score based on the list of conditions mentioned in the updated review #1

[Author Response · NeurIPS 2020]

| $\varepsilon$ | synthetic-RMSE | | realworld-RMSE | |
|---|---|---|---|---|
| | CGLSE | Uni | CGLSE | Uni |
| 0.1 | **.005** | .007 | **.008** | .014 |
| 0.2 | **.011** | .017 | **.021** | .029 |
| 0.3 | **.022** | .026 | **.065** | .073 |
| 0.4 | **.032** | .056 | **.116** | .136 |
| 0.5 | **.051** | .099 | **.167** | .207 |

Table 1: existing simulations - RMSE

| $\varepsilon$ | max. emp. err. | | avg. (std.) of emp. err. | | RMSE | | size |
|---|---|---|---|---|---|---|---|
| | CGLSE | Uni | CGLSE | Uni | CGLSE | Uni | |
| 0.1 | .029 | 1.574 | .026(.002) | .984(.305) | **.026** | 1.030 | 68220 |
| 0.2 | .042 | 1.465 | .015(.011) | 1.066(.206) | **.019** | 1.085 | 10977 |
| 0.3 | .046 | .998 | .023(.012) | .976(.012) | **.026** | .976 | 3356 |
| 0.4 | .088 | .997 | .055(.017) | .995(.057) | **.057** | .995 | 1428 |
| 0.5 | .071 | .996 | .067(.003) | .985(.067) | **.067** | .985 | 741 |

Table 2: new simulations - synthetic data (Cauchy)

We sincerely thank all the reviewers for their insightful comments. We take these comments seriously and address them below. This paper was first submitted to ICML 2020. The paper got **one accept and two weak accepts**. The main comments were to focus on a single regression problem (GLSE) in the main body of the paper and include empirical results with real world data – we did both. In doing so – we moved $\text{GLSE}_k$ to the appendix – this seems to have diluted the technical novelty of the current paper for some of the current reviewers. While this is disappointing, we fully understand – we can easily bring back the emphasis on $\text{GLSE}_k$ where there is significant technical novelty.

**Technical novelty for $\text{GLSE}_k$. Novelty 1: Coreset definition.** The first difficulty is that, unlike GLSE, due to the $\min$ operation, the objective function $\psi^{(G,q,k)}$ of $\text{GLSE}_k$ can only be decomposed into sub-functions $\psi_i^{(G,q,k)}$ instead of individual-time pairs. We address this by incorporating $\min$ operations in the computation function $\psi_S^{(G,q,k)}$ over the coreset $S$. The second difficulty is that the clustering centers are *subspaces* induced by regression vectors $\beta^{(1)}, \ldots, \beta^{(k)}$, instead of *points* as in Gaussian mixture models or $k$-means. So it is unclear how $\text{GLSE}_k$ can be reduced to projective clustering used in Gaussian mixture models (see also [Feldman et al., 2019, Coresets for Gaussian mixture models of any shape]). We address this by treating observation vectors of an individual $(x_{i1}, \ldots, x_{iT})$ as one entity while constructing coresets. **Novelty 2: Coreset construction/upper bounding total sensitivity.** This involves two steps. In **Step 1**, we reduce the sensitivity function from $\text{GLSE}_k$ to $\text{OLSE}_k$ (Lemma D.10), based on two observations: for any $\zeta = (\beta, \rho) \in \mathcal{P}_\lambda$ (recall that $\mathcal{P}_\lambda = \mathbb{R}^d \times B_{1-\lambda}^q$ for some constant $\lambda \in (0,1)$ where $B_{1-\lambda}^q = \{\rho \in \mathbb{R}^q : \|\rho\|_2^2 \leq 1 - \lambda\}$) we have $\psi_i^{(G,q)}(\zeta) \geq \lambda \cdot \psi_i^{(O)}(\beta)$ that provides an upper bound of the individual objective gap between GLSE and OLSE, and for any $\zeta = (\beta^{(1)}, \ldots, \beta^{(k)}, \rho^{(1)}, \ldots, \rho^{(k)}) \in \mathcal{P}^k, \psi_i^{(G,q,k)}(\zeta) \leq 2(q+1) \cdot \min_{l \in [k]} \psi_i^{(O)}(\beta^{(l)})$; and for any $\zeta = (\beta^{(1)}, \ldots, \beta^{(k)}, \rho^{(1)}, \ldots, \rho^{(k)}) \in \mathcal{P}_\lambda^k, \psi_i^{(G,q,k)}(\zeta) \leq 2(q+1) \cdot \min_{l \in [k]} \psi_i^{(O)}(\beta^{(l)})$, that provides a lower bound of the individual objective gap between $\text{GLSE}_k$ and $\text{OLSE}_k$. **Step 2** upper bounds the total sensitivity of $\text{OLSE}_k$. This key step for coreset construction (Lines 3-4 in Algorithm 2) is done by showing that the max. influence of individual $i$ is at most $\frac{u_i}{u_i + \sum_{j \neq i} \ell_j}$ where $u_i$ is the largest eigenvalue of $(Z^{(i)})^\top Z^{(i)}$ and $\ell_j$ is the smallest eigenvalue of $(Z^{(j)})^\top Z^{(j)}$, where $Z^{(i)} \in \mathbb{R}^{T \times (d+1)}$ is the matrix whose $t$-th row is $z_t^{(i)} = (x_{it}, y_{it}) \in \mathbb{R}^{d+1}$ (Definition D.3 and Lemma D.9).

**Empirical performance of our coresets.** Reviewers note weak performance of coresets relative to uniform on average error. As noted by R1, uniform may have lower average error, but has higher std (more large errors), i.e., many observations may be poorly represented. Std is also higher in real data, where errors may not be "regular" as in Gaussian noise. To illustrate this, we will include RMSE (root mean square error)–a standard metric of performance. Given a set of errors $e_1, \ldots, e_n$, **RMSE** $:= \sqrt{\frac{1}{n} \sum_{i \in [n]} e_i^2}$. In Table 1, RMSE(coresets) < RMSE(uniform) in both datasets—60%-90% of uniform on real data and 50%-85% of uniform on synthetic data with Gaussian errors. Further, unlike coresets, uniform also has no performance bounds on max error. For instance, in the real-world data with $\varepsilon = 0.5$, the max. error of uniform is .775 which exceeds the desired error bound. R1 asks about role of leverage. Coresets should perform better with high leverage observations, given max. error guarantees. We earlier presented Gaussian error (low leverage, few outliers) because it is a "hard" benchmark to beat. We illustrate this by replacing Gaussian errors in Eq. (2) with the **Cauchy** (0,2) distribution that has heavy tails as in $e_{it} = \sum_{a=1}^{\min\{t-1,q\}} \rho_a e_{i,t-a} + \textbf{Cauchy}(0,2)$. Table 2 shows that coreset performance relative to uniform is now even better for max/avg/RMSE errors. The max error for uniform exceeds the desired bound for all values of $\varepsilon$; and is at least 10x that of our coreset. In summary, we greatly appreciate the issues raised. We note that the issues raised (performance on real data, outliers/leverage points) strengthen evidence in favor of our coreset. We will add these points to the final version.

**R1.** Thank you for your constructive feedback. The final version will account for your expositional suggestions. Hopefully we have addressed your concerns and we hope you will support our paper.

**R2.** Thank you for appreciating our paper, providing positive feedback, and supporting it. The code is already on github (not included for anonymity); a link will be added in final version.

**R3.** Thanks for appreciating the novelty of our GLSE result. We hope the above discussion on empirical results and technical novelty (for $\text{GLSE}_k$) addresses your concerns. We hope you will strengthen support for our paper.

**R4.** Thank you for your detailed feedback. We clarify that our GLSE coreset with AR($q$) **works for any** $q' \leq q$ and any $\rho \in B^q$. On the real-world use of $\text{GLSE}_k$, it is a basic problem with applications in many fields; as accounting for *unobserved heterogeneity* in panel regressions is critical for unbiased estimates. See, Arellano, M. (2003). *Panel data econometrics*. Halaby, C. N. (2004). *Panel models in sociological research*. Annual Review of Sociology. We will add a discussion on the importance of $\text{GLSE}_k$. We will add boxplots in the full version. We hope you will support our paper.



[Meta-Review · NeurIPS 2020]

This paper provides a coreset method for a generalization of ordinary least squares regression. The reviewers seem to be in agreement that, especially when considering work in the supplement, the approach is interesting and has the potential for impact due to the usefulness of the least squares generalization but also by enabling yet further future extensions. The authors seem to have already agreed to a number of useful changes for the camera ready, and I will just emphasize a couple here: * Moving up GLSE_k to the main text (and discussing novelty) * Incorporating new experiments I also want to point to the detailed list of conditions in R1's updated review (post-author-response). The authors should be careful to read these and address them as best they can.